# KoLA: Carefully Benchmarking World Knowledge of Large Language Models

**Jifan Yu,**[*] **Xiaozhi Wang,**[*] **Shangqing Tu,**[*] **Shulin Cao, Daniel Zhang-Li, Xin Lv,**
**Hao Peng, Zijun Yao, Xiaohan Zhang, Hanming Li, Chunyang Li,**
**Zheyuan Zhang, Yushi Bai, Yantao Liu, Amy Xin, Nianyi Lin, Kaifeng Yun,**
**Linlu Gong, Jianhui Chen, Zhili Wu, Yunjia Qi, Weikai Li, Yong Guan,**
**Kaisheng Zeng, Ji Qi, Hailong Jin, Jinxin Liu, Yu Gu, Yuan Yao, Ning Ding,**
**Lei Hou, Zhiyuan Liu, Bin Xu, Jie Tang, Juanzi Li**[†]
Tsinghua University, Beijing, China, 100084
`kola-benchmark@googlegroups.com`

## Abstract

The unprecedented performance of large language models (LLMs) necessitates improvements in evaluations. Rather than merely exploring the breadth of LLM abilities, we believe meticulous and thoughtful designs are essential to thorough, unbiased, and applicable evaluations. Given the importance of world knowledge to LLMs, we construct a Knowledge-oriented LLM Assessment benchmark (KoLA), in which we carefully design three crucial factors: (1) For **ability modeling**, we mimic human cognition to form a four-level taxonomy of knowledge-related abilities, covering 19 tasks. (2) For **data**, to ensure fair comparisons, we use both Wikipedia, a corpus prevalently pre-trained by LLMs, along with continuously collected emerging corpora, aiming to evaluate the capacity to handle unseen data and evolving knowledge. (3) For **evaluation criteria**, we adopt a contrastive system, including overall standard scores for better numerical comparability across tasks and models and a unique self-contrast metric for automatically evaluating knowledge-creating ability. We evaluate 28 open-source and commercial LLMs and obtain some intriguing findings. The KoLA dataset will be updated every three months to provide timely references for developing LLMs and knowledge systems.

## 1 Introduction

Recent remarkable breakthroughs achieved by large language models (LLMs) like GPT-4 (OpenAI, 2023) have elicited widespread astonishment. Considering the extensive and profound natural language understanding and generation abilities exhibited by LLMs (Bubeck et al., 2023), the conventional benchmarks (Wang et al., 2018; 2019) focusing on relatively narrow and superficial abilities are no longer as helpful for testing them. It has become necessary to construct better benchmarks for effectively comparing LLMs and providing valuable diagnostic results. To this end, various benchmarks are proposed, focusing on extending the evaluation scope to cover broader abilities (Hendrycks et al., 2021; Zhong et al., 2023; Huang et al., 2023) or more challenging tasks (Srivastava et al., 2022; Suzgun et al., 2022).

In addition to broadening the evaluation scope to explore the breadth of LLM abilities, we believe meticulous designs are also necessary to build evaluations that facilitate in-depth insights, maintain impartiality towards different LLMs, and have high applicability for audiences interested in selecting and enhancing LLMs. Designing a benchmark requires careful consideration of three key factors: (1) **Ability Modeling**. A benchmark should not only define the scope of desired abilities but also model the inherent connections between the evaluated abilities, which allows for diagnostic insights on how to acquire and improve these abilities. (2) **Data**. Given the extremely broad range of training data for LLMs, which might include annotated data of certain tasks and is often undisclosed, ensuring

---

[*]Equal Contribution.
[†]Corresponding author.

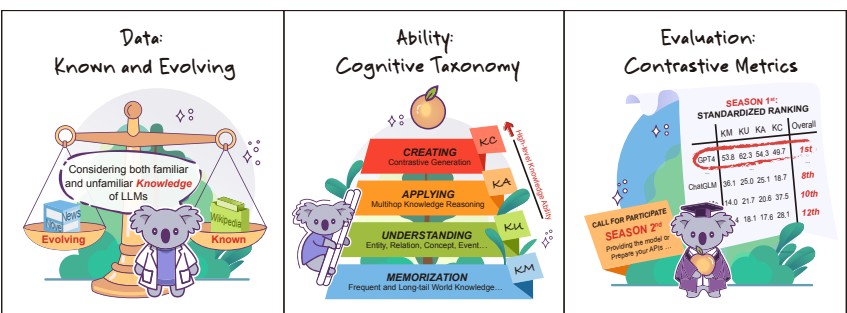

Figure 1: KoLA's careful design on three key factors for LLM evaluation.

that differences in training data do not impact the evaluation fairness is critical and challenging. (3) **Evaluation Criteria**. For high applicability, evaluation metrics should enable audiences to easily understand and gain helpful observations. Moreover, there are many well-known issues (Theis et al., 2016; Sajjadi et al., 2018; Ji et al., 2023) for evaluating tasks with large search spaces like the generative tasks. Evaluations for related abilities still heavily rely on human evaluation, which is time-consuming and not easily reproducible (Belz et al., 2022; 2023).

In this paper, we propose a Knowledge-oriented LLM Assessment benchmark (KoLA), which aims at carefully benchmarking the world knowledge of LLMs by undertaking meticulous designs considering the aforementioned three factors:

For ability modeling, we evaluate world knowledge of LLMs and design a **four-level cognitive ability taxonomy**. We chose world knowledge as our evaluation scope because: (i) World Knowledge is widely recognized as playing a fundamental role in the impressive performance of LLMs (Hendrycks et al., 2021; Zhong et al., 2023; Huang et al., 2023), and a deeper grasp of knowledge enables LLMs to better assist humans; (ii) Recent work has shown that understanding and generating structural world knowledge remain challenging for LLMs. Unlike previous work focusing on expanding the evaluation breadth by covering diver tasks and disciplinary knowledge to test the knowledge boundaries of LLMs (Hendrycks et al., 2021; Zhong et al., 2023; Huang et al., 2023), we focus more on the "depth" of evaluation, i.e., modeling the intrinsic connections between knowledge-related abilities and ensuring reliable evaluation results. Inspired by the human cognitive processes in learning theory, such as Bloom's taxonomy (Krathwohl, 2002), we organize evaluated abilities into four levels: Knowledge Memorization, Knowledge Understanding, Knowledge Applying, and Knowledge Creating. This taxonomy helps to provide more specific and helpful evaluation results, detailing which aspect of knowledge the evaluated models may be deficient in. It also facilitates a preliminary exploration of the similarities and differences between the learning mechanisms of LLMs and humans. To coordinate with our data design considerations introduced later, we selected 19 tasks, primarily focusing on world knowledge about entities, concepts, and events.

For data, we obtain both **known and evolving data sources**. Some studies adopt unpublished or machine-unreadable data (Zhong et al., 2023; Huang et al., 2023) to reduce the possibility that the test data has been learned by LLMs. However, considering the intense competition between LLMs, those data may also be trained by LLMs in the near future[1]. We believe the ideal approach is to do evaluations on newly emerging data and maintain a continuously evolving benchmark, like the attempts that include time-sensitive evolving data (Kasai et al., 2022; Dhingra et al., 2022). In KoLA, we host a new competition season every three months. For each season, we crawl and annotate 500 recently published articles as the evolving data. The evolving data source allows us to (i) evaluate models more fairly, even if some models can rapidly update their knowledge, thereby demonstrating their power, and (ii) better track the model development. Besides evolving data, we also consider the known data of LLMs, which means the data sources that all models have learned. Evaluations on known data enable us to (i) fairly compare the learning efficiency of LLMs by comparing the different knowledge they acquire from the same training data and (ii) assess the generalization ability by comparing LLMs' performance on known data and evolving data. We chose Wikipedia as our known data source due to its common use. Considering the limitations of Wikipedia and our annotation capabilities on the evolving data, we are unable to cover a very wide range of tasks.

---

[1]Some cases have been noted (https://cevalbenchmark.com/static/leaderboard.html)

For evaluation criteria, we design a **contrastive evaluation system**, including an overall standard score system and a self-contrast knowledge creating metric. Conventional benchmarks report absolute metrics for different tasks separately. The incomparability of scores across tasks makes it difficult for audiences to intuitively compare the proficiency levels across different abilities. Additionally, the sensitivity of different metrics varies, which may lead less experienced audiences to misinterpret the ability differences represented by numerical differences. In the KoLA main leaderboard, we report standard scores across different tasks, determined by the relative level compared to other evaluated LLMs. This makes KoLA applicable to a broader range of audiences. Experienced audiences can still refer to absolute metrics if desired. Furthermore, evaluating knowledge creation is particularly challenging as it involves distinguishing the correctly created knowledge and knowledge hallucinations (Ji et al., 2023). We design a self-contrast metric for evaluating knowledge hallucination by contrasting freely created completions and knowledge-grounded completions of an LLM given the same beginnings. This metric eliminates the influence of writing styles and focuses on whether the generated completions are consistent with the actually presented knowledge.

In the first two seasons of KoLA, we evaluate 28 widely-used LLMs, including 8 API-access commercial LLMs, such as GPT-4 (OpenAI, 2023) and Cohere-command, and 20 open-source LLMs including GLM-130B (Zeng et al., 2022), LLaMa (Touvron et al., 2023), etc. From the experimental results, we obtain some intriguing observations, such as larger base models tend to memorize more knowledge, alignment unleashes the potential of larger models in higher-level abilities but may harm memorization, and open-source models exhibit overall inferiority compared to commercial models.

We welcome the participation of more LLMs in KoLA evaluation and encourage contributions to the new seasons of KoLA. The data, leaderboard, participation information, and supporting tools are publicly available upon acceptance. We hope KoLA can serve as a diagnostic tool to facilitate the development of increasingly knowledgeable LLMs, and also help practitioners select LLMs.

## 2 KoLA Benchmark

### 2.1 Ability Modeling

Within the context of Artificial Intelligence (AI), *Knowledge* has long been employed to signify *information encompassing facts, events, and skills* (Feigenbaum, 1977), serving as an indicator for the intelligence level of AI. Hence various *knowledge-intensive tasks* (Petroni et al., 2019; 2021) are proposed to examine language models' knowledge-related abilities. Recently, the impressive performance of LLMs has encouraged the development of more comprehensive benchmarks (Srivastava et al., 2022; Suzgun et al., 2022) with broad human-subject exams (Hendrycks et al., 2021; Zhong et al., 2023; Huang et al., 2023).

**Cognitive Ability Taxonomy**. Confronted with such a vast array of evaluation datasets, we advocate for considering the stratification and connection of abilities, rather than organizing them discretely (Wang et al., 2019; Hendrycks et al., 2021; Srivastava et al., 2022; Suzgun et al., 2022) or straightforwardly based on disciplines (Zhong et al., 2023) or difficulties (Huang et al., 2023). Such viewpoints have also been upheld by cognitive scientists for several decades, giving rise to a series of cognitive learning theories (Lewis & Smith, 1993). Considering the ongoing debates surrounding high-order thinking (Miri et al., 2007; Collins, 2014), we simplify and select four widely accepted processes in Bloom's taxonomy (Krathwohl, 2002) for organizing the tasks in KoLA benchmark.

1. **Knowledge Memorization (KM)** aims to gauge the model's ability in faithfully recalling known facts, exemplified by the previous knowledge probing task (Petroni et al., 2019).

2. **Knowledge Understanding (KU)** focuses on evaluating the model's ability in understanding the underlying knowledge within texts, instantiated by the conventional information extraction tasks (Yao et al., 2019; Wang et al., 2020; Ding et al., 2021; Peng et al., 2022).

3. **Knowledge Applying (KA)** reflects the ability of agents in employing knowledge to accomplish reasoning and problem-solving tasks. Consequently, this level is evaluated by various knowledge reasoning tasks (Yang et al., 2018; Trivedi et al., 2022; Cao et al., 2022).

4. **Knowledge Creating (KC)** denotes the ability to create novel and reasonable knowledge given known facts. This is evaluated by the knowledge coherence and correctness (Chen

et al., 2020; Bang et al., 2023) of contents generated by the model. It is worth noting that the evaluation goes beyond merely assessing the generation quality (fluency, etc.).

## 2.2 DATA SOURCE AND SELECTED TASKS

**Known & Evolving Data**: A common concern in evaluating LLMs is the fairness issue brought by variations in training data and the potential test data leakage risk. To minimize these biases, we propose the design of the following distinctive data sources:

(1) *Known Data Source*. Wikipedia[2] is an acknowledged high-quality corpus containing over 6.6 million English articles, which has been used in pre-training by numerous pre-trained models since BERT (Devlin et al., 2019; Brown et al., 2020; Shuster et al., 2022) and is widely included in open pre-training corpora (Gao et al., 2021). Hence we believe assuming every LLM has been trained on Wikipedia is reasonable and adopt it as our known data source. Considering that many LLMs state they can only provide answers based on "Content before 2021" [3], we select Wikidata5M (Wang et al., 2021a), a high-quality subset of Wikidata, as the basis, which allows linking to the 2019 version of Wikipedia dump, thus enabling the selection or reconstruction of downstream tasks' datasets.

(2) *Evolving Data Source*. Considering the time required for model training (Zeng et al., 2022), it is less unlikely for newly emerged data to be timely trained by LLMs. Therefore, we have devised an evolving evaluation mechanism that continuously retrieves the web content published in around recent 90 days as the data source and constructs new datasets on them. This approach ensures fair assessment of LLMs' performance on unseen content and whether they "secretly" involve knowledge updating modules like the external search. Each update (we call it a *Season* of KoLA) requires crawling a minimum of 500 articles to support building test sets. For the first season reported in this paper, we adopt two kinds of data: factual news [4] and fictional novels [5]. We intend to persist for an additional 4 seasons (approximately 1 year) to promptly integrate the forthcoming top LLMs. We anticipate that the consistently released reports can further support relevant researchers.

Built upon these two data sources, we finally select and construct 19 tasks in KoLA, as shown in Table 1. To ensure both the quality and efficiency of annotations for each season, we randomly select one task at each level to annotate the new evolving evaluation dataset. For the existing datasets, we try to ensure most of the test sets are not public, and this rigorous setting ensures a high level of fairness. The data collection and task construction details are shown in Appendix C. We briefly introduce the tasks of the four levels below. It is noteworthy that, due to the limitations of data distribution and collection processes, the absolute numerical values of the model on *Evolving* data are not necessarily destined to be lower than those on *Known* data.

**Knowledge Memorization Tasks**: We follow LAMA (Petroni et al., 2019) to evaluate knowledge memorization by probing facts from LLMs but re-construct the datasets on our data sources. Given a triplet in Wikidata5M (Wang et al., 2021a), we transform it into a sentence with a relation-specific template and let LLMs complete its tail entity. Additionally, we want to explore whether the knowledge memorization of LLMs correlates with training frequency. We sort the entities in Wikidata5M according to their frequency of occurrence in Wikipedia (Jin et al., 2019), resulting in the creation of two test sets: (1-1) *High-Frequency Knowledge*. Randomly selecting 100 entities from the top $2,000$ entities with the highest frequency and construct data with triplets of them; (1-2) *Low-Frequency Knowledge*. Similarly, we randomly select 100 entities from the lowest-frequency entities and construct a more challenging evaluation set; (1-3) *Evolving Test of Memorization (ETM)*. From the articles in evolving data sources, we annotate the knowledge triplets shown in them and only preserve 100 triplets that cannot be inferred from previously available corpora.

**Knowledge Understanding Tasks**: Knowledge understanding is evaluated by whether LLMs can understand various genres of knowledge from texts, including concepts, entities, entity relations, events, and event relations. (2-1/2-2/2-3) *Concept Probing* employs the three probing tasks (CSJ, CPJ, CiC) of COPEN (Peng et al., 2022) to evaluate the models' understanding of conceptual knowledge. (2-4) *Named Entity Recognition* utilizes the FewNERD dataset (Ding et al., 2021), from which we

---

[2]https://www.wikipedia.org

[3]https://chat.openai.com

[4]An open source news API at Github. URL: https://github.com/ranahaani/GNews

[5]A well-known open license novel creating community. URL: https://archiveofourown.org

Table 1: The tasks in KoLA (Season 1st and 2nd). Metrics in bold are selected for calculating standardized scores. *Exclusive* task means their test sets are newly developed or sponsored by the original authors and were not publicly disclosed. Test Set and Pool correspond to the testing instances used in each season and the overall available instances.

| Level | ID | Dataset | Metrics | Exclusive | Context Type | Test Set | Pool | Source |
|---|---|---|---|---|---|---|---|---|
| KM | 1-1 | High-Freq. | EM, **F1** | ✔ | Triple | 100 | 20.6M | Known |
| | 1-2 | Low-Freq. | EM, **F1** | ✔ | Triple | 100 | 20.6M | |
| | 1-3 | ETM | EM, **F1** | ✔ | Triple | 100 | 2.7k | Evolving |
| KU | 2-1 | COPEN-CSJ | **Acc.** | ✔ | Entity, Concept | 100 | 3.9k | Known |
| | 2-2 | COPEN-CPJ | **Acc.** | ✔ | Concept | 100 | 4.7k | |
| | 2-3 | COPEN-CiC | **Acc.** | ✔ | Concept | 100 | 2.3k | |
| | 2-4 | FewNERD | **F1** | ✘ | Sentence | 300 | 188.2k | |
| | 2-5 | DocRED | **F1** | ✔ | Document, Entity | 100 | 12k | |
| | 2-6 | MAVEN | **F1** | ✔ | Document | 100 | 20.4k | |
| | 2-7 | MAVEN-ERE | **F1** | ✔ | Document(s), Event | 199 | 1.3M | |
| | 2-8 | ETU | **F1** | ✔ | Document, Entity | 100 | 1.6k | Evolving |
| KA | 3-1 | HotpotQA | **F1** | ✘ | Document(s) | 100 | 7.4k | Known |
| | 3-2 | 2WikiMulti. | **F1** | ✔ | Document(s) | 100 | 12.6k | |
| | 3-3 | MuSiQue | **F1** | ✔ | Document(s) | 100 | 2.5k | |
| | 3-4 | KQA Pro | **F1** | ✔ | KG | 100 | 1.2k | |
| | 3-5 | KoRC | **F1** | ✔ | Document(s), KG | 100 | 5.2k | |
| | 3-6 | ETA | **F1** | ✔ | Document(s), KG | 49 | 1.6k | Evolving |
| KC | 4-1 | Encyclopedic | BLEU, **Rouge** | ✔ | Document, Event | 95 | 4.5k | Known |
| | 4-2 | ETC | BLEU, **Rouge** | ✔ | Document, Event | 95 | 100 | Evolving |

randomly select 300 examples in our evaluation. (2-5) *Relation Extraction* selects the undisclosed test set from the challenging document-level relation extraction dataset, DocRED (Yao et al., 2019). (2-6) *Event Detection* adopts the undisclosed test set of the finely annotated MAVEN (Wang et al., 2020) dataset. (2-7) *Event Relation Extraction* involves the undisclosed test set from MAVEN-ERE (Wang et al., 2022), which consists of 113k examples of coreference, temporal, causal, and subevent relations between events. (2-8) *Evolving Test of Understanding (ETU)*. For the articles in evolving data, we conduct the entity recognition and follow the same relation schema of DocRED to annotate a brand new test set containing 100 relation instances from 50 articles. It is worth noting that apart from the evolving test, the other datasets are all based on Wikipedia texts.

**Knowledge Applying Tasks**: Knowledge applying ability is evaluated by LLMs' multi-hop reasoning capabilities, specifically over world knowledge. This differs from several recent studies (Lu et al., 2023; Mialon et al., 2023), which cover more general reasoning, such as mathematical reasoning. Therefore, the following progressive Wikipedia-based datasets are included in KoLA: (3-1) *HotpotQA* (Yang et al., 2018) is a question-answering dataset that involves a substantial number of natural language questions written by native speakers, examining machine's abilities in comparison, multi-hop reasoning, and more. However, a limitation of HotpotQA is that some questions can be answered through shortcuts. To address this, (3-2) *2WikiMultihopQA* (Ho et al., 2020) ensures that questions cannot be solved through shortcuts by manually-designed templates, but their questions lack naturalness in language. Furthermore, the (3-3) *MuSiQue* (Trivedi et al., 2022) dataset tackles the challenges of shortcuts and naturalness simultaneously. Its questions are composed of simple questions from existing datasets, with up to four-hop complex reasoning. (3-4) *KQA Pro* (Cao et al., 2022) is a large-scale dataset, whose questions are relatively complex, allowing for more fine-grained evaluation of LLMs' multi-hop reasoning with logical operations and modifiers. (3-5) *KoRC* (Yao et al., 2023) is a dataset that requires joint reasoning between the text and knowledge base. It differs from the aforementioned four datasets as it requires implicit rather than explicit reasoning. (3-6) *Evolving Test of Applying (ETA)* takes the same construction approach as KoRC, producing 49 questions upon 350 annotated knowledge triplets and 40 articles in the evolving data.

**Knowledge Creating Tasks**: As the highest level of *Bloom's Cognitive Taxonomy* (Krathwohl, 2002), how to evaluate knowledge creation is a long-standing open and challenging question. The capacity for knowledge creation is evident in open-ended generation tasks. Traditional text generation evaluation metrics (Theis et al., 2016) are based on textual similarities between model-generated content and human-written references, which do not solely focus on knowledge creation ability but

cover other skills, such as text style and fluency (Naeem et al., 2020). Ideally, human evaluators should be employed to solely assess whether the content generated by models contains novel and reasonable knowledge (Maher, 2010; Lamb et al., 2018). However, manually evaluating diverse open-domain knowledge is labor-intensive, costly, and lacks scalability. Inspired by the knowledge-grounded text generation tasks (Yu et al., 2022), KoLA proposes a feasible automatic evaluation protocol that specifically contrasts the model-generated knowledge with that in human references. First, we limit the generation scope to *narrative texts* such as history, news, and fiction. This is because the knowledge creating in generating narrative texts has a clear focus on envisioning plausible subsequent *events* and articulating them in a reasonable way. As shown in Figure 2, we then conduct human annotation on the reference texts to obtain reference fine-grained event knowledge. The annotated events enable a dedicated self-contrast metric (elaborated below) that emphasizes the quality of event knowledge in generated content. This approach offers an effective assessment of knowledge creation abilities compared to traditional text generation metrics encompassing many other factors (Akter et al., 2022). We conduct annotation on both Wikipedia texts and evolving articles, which constructs two evaluation datasets: (4-1) *Encyclopedic Knowledge Creation*, which is based on the narrative Wikipedia articles selected by MAVEN (Wang et al., 2020) and (4-2) *Open Knowledge Creation*, which is based on unseen news and novels, serving as the *Evolving Test of Creating (ETC)*.

Table 1 presents the features and statistics of each selected task. Further details regarding annotation processes and task demonstrations are correspondingly presented in Appendix D.

## 2.3 CONTRASTIVE EVALUATION SYSTEM

Our contrastive evaluation system includes standardized overall scores based on relative model comparisons and a unique self-contrast metric, which can automatically evaluate knowledge hallucination and enhance generation evaluation.

**Standardized Overall Scoring**. Since the metrics of different KoLA tasks are incomparable and differently sensitive, less experienced audiences cannot easily compare and interpret results, which is also prevalent in recent LLM benchmarks like Big-Bench-Hard (Suzgun et al., 2022) and MMLU (Hendrycks et al., 2021). Therefore, we propose to introduce standardized scores (Dyck et al., 2005) to enhance the applicability of KoLA results. Specifically, given a task set $D = \{d_i\}_{i=1}^{|D|}$ and the evaluated model set $M = \{m_j\}_{j=1}^{|M|}$, we first select the most representative metric for each task, allowing us to compute the performance score $x_{ij}$ of model $m_j$ on task $d_i$. Then the standardized score $z$ can be calculated as:

$$z_{ij} = \frac{x_{ij} - \mu\left(x_{i1}, ..., x_{i|M|}\right)}{\sigma\left(x_{i1}, ..., x_{i|M|}\right)}, \tag{1}$$

where $\mu\left(\cdot\right)$ and $\sigma\left(\cdot\right)$ denote the mean and standard deviation. Subsequently, we apply Min-Max scaling (Patro & Sahu, 2015) to adjust all the results to the range of $[0, 100]$, further enhancing the correlation and readability of scores across tasks. The final scores are presented as:

$$s_{ij} = 100\frac{z_{ij} - \mathtt{min}\left(z\right)}{\mathtt{max}\left(z\right) - \mathtt{min}\left(z\right)}, \tag{2}$$

where the functions $\mathtt{max}\left(z\right)$ and $\mathtt{min}\left(z\right)$ correspond to the maximum and minimum of all $z_{ij}$ scores.

**Self-contrast Metric**. Evaluating knowledge creating is not only about evaluating generation quality (Theis et al., 2016), but more about assessing whether the generated knowledge is faithful and reasonable, i.e., avoiding *knowledge hallucination* (Ji et al., 2023). We develop a unique self-contrast metric for this, which is defined by contrasting two completions generated by the same model.

As illustrated in Figure 2, $C$ denotes the given preceding context, $R$ denotes the human-written succeeding completion, and $K$ refers to the annotated event knowledge in $R$. Each model is required to generate two completions: (a) Given only context $C$, generate a version of completion $T$, which requires the model to freely imagine the possible events and may have knowledge hallucination like the negotiation event in Figure 2; (b) Given both the context $C$ and the *foreknowledge* $K$, generate another completion $T_k$, which only requires the model to reasonably compose the given events. If $T$ and $T_k$ demonstrate a strong resemblance, it implies that the model can create highly reasonable events that are consistent with human-provided references and has less knowledge hallucination. The distinct advantage of this self-contrast method is that, since both completions are generated by the same model, factors external to knowledge creation, such as writing style, are highly likely to remain consistent,

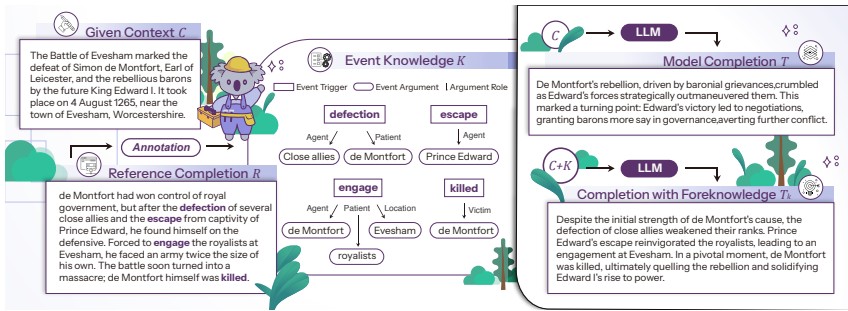

Figure 2: Illustration of the Knowledge Creating (KC) tasks.

minimizing their influence on the evaluation. Moreover, to cover more comprehensive aspects of knowledge creation abilities and prevent evaluation collapse caused by the model's disregard for the knowledge $K$ in the prompt of the process (b), the overall knowledge creating score is defined as the mixture of multiple contrasts:

$$x = \mathtt{avg}\left(\partial\left(T, R\right), \partial\left(T, T_k\right), \partial\left(T_k, R\right)\right), \tag{3}$$

where $\mathtt{avg}\left(\cdot\right)$ denotes the average. Function $\partial\left(\cdot\right)$ is to calculate the similarity of the two texts, which we employ the widely-used Rouge-L (F1) (Lin, 2004) in this work. The $\partial\left(T, R\right)$ is the conventional text generation metric. While it captures a broad range of knowledge creation abilities (spanning multiple genres of knowledge beyond events), it also includes undesired factors unrelated to knowledge creation, such as writing styles and text fluency. Hence we add $\partial\left(T, T_k\right)$ and $\partial\left(T_k, R\right)$ to emphasize the abilities about creating event-related knowledge, which is important for generating narrative texts. The $\partial\left(T, T_k\right)$ is the newly-proposed self-contrast metric focusing on whether the generated event knowledge is reasonable. The $\partial\left(T_k, R\right)$ takes inspiration from the knowledge-grounded generation tasks (Chen et al., 2020; Ghazvininejad et al., 2018). It reflects the ability to create knowledge about the relationships between events, which is required in reasonably composing the given events into a story. For instance, the $T_k$ in Figure 2 implies that the death of Simon de Montfort caused the rebels to lose the battle, while this is a hallucinated causal relation inconsistent with the narrative in $R$.

## 3 EXPERIMENT

**Evaluated Models.** In the first two seasons of KoLA, we evaluate LLMs of two categories: (1) *Open-source Model*, including GPT-J (6B) (Wang & Komatsuzaki, 2021), GPT-JT (6B) (Computer, 2022), GPT-NeoX (20B) (Black et al., 2022), BLOOM (7B) (Scao et al., 2022), T0++ (11B) (Bang et al., 2023), LLaMa (65B) (Touvron et al., 2023), GLM (130B) (Zeng et al., 2022), UL2 (20B) (Tay et al., 2022), FLAN-T5 (11B) (Chung et al., 2022), FLAN-UL2 (20B) (Research, 2022), Alpaca (7B) (Taori et al., 2023), ChatGLM (6B) (Du et al., 2022), Dolly-v2 (12B) (Conover et al., 2023),RedPajama-Instruct (7B) (Computer, 2023), Tulu (7B) (Wang et al., 2023), Vicuna (13B) (Zheng et al., 2023), Llama2-chat (7B) (Touvron et al., 2023), ChatGLM2-32k (6B) (Zeng et al., 2022), Internlm-chat-8k (7B) (Team, 2023); (2) *API service*: GPT-3 curie v1 (6.7B)[6] and davinci v1 (175B) (Brown et al., 2020), InstructGPT curie v1 (6.7B*)[6] and davinci v2 (175B*) (Ouyang et al., 2022), ChatGLM (130B) (Zeng et al., 2022), Cohere-command (52.4B)[7], J2-Jumbo-Instruct (178B*) (Studio, 2023), GPT3.5-turbo[6] and GPT-4 (OpenAI, 2023). (*) indicates the size has not been confirmed.

**Overall Performance.** We report the standardized scores of all models in Table 2 and 3, where "—" indicates that the result is unavailable due to the input is longer than the model context length. All results are from **Season 2nd** (Sept. 2023), and the comparison with the rankings from Season 1st (June 2023, Appendix F) is shown in the "Rank" column. Despite the overall consistency in rankings across different levels, we can still obtain some intriguing findings from the results:

(1) For models without alignment or instruction tuning (e.g., GPT-J and BLOOM), there is a strong correlation (Spearman's coefficient of 0.79) between the ranking of the Knowledge Memory (KM)

---

[6] https://platform.openai.com/overview
[7] https://docs.cohere.com/docs/the-command-model

Table 2: Standardized performance of Knowledge Memorization and Understanding level.

| Model | Level 1: KM | | | | Level 2: KU | | | | | | | | |
|---|---|---|---|---|---|---|---|---|---|---|---|---|---|
| | 1-1 | 1-2 | 1-3 | Rank | 2-1 | 2-2 | 2-3 | 2-4 | 2-5 | 2-6 | 2-7 | 2-8 | Rank |
| GPT-4 | 64.3 | 68.9 | 41.5 | 1st (—) | 69.5 | 48.6 | 51.4 | 66.9 | 100.0 | 77.2 | 78.9 | 81.3 | 1st (—) |
| GPT-3.5-turbo | 53.4 | 60.0 | 38.3 | 2nd (↑2) | 44.2 | 49.4 | 50.2 | 54.0 | 51.3 | 50.2 | 56.6 | 25.5 | 2nd (—) |
| InstructGPT davinci v2 (175B*) | 41.2 | 48.4 | 36.1 | 7th (↑1) | 33.6 | 48.2 | 42.4 | 41.8 | 56.7 | 62.3 | 40.6 | 31.3 | 3rd (—) |
| Tulu (7B) | 41.5 | 48.6 | 28.4 | 8th (↓2) | 22.0 | 25.4 | 43.0 | 35.7 | 30.9 | 20.6 | 22.2 | 25.8 | 11th (↑1) |
| Cohere-command (52.4B) | 59.0 | 54.5 | 33.9 | 3rd (↓1) | 40.0 | 47.1 | 46.3 | 26.6 | 38.6 | 20.6 | 46.9 | 25.0 | 4th (—) |
| FLAN-UL2 (20B) | 53.0 | 42.6 | 30.7 | 6th (↓1) | 59.0 | 47.1 | 53.0 | 16.0 | 25.0 | 20.6 | 22.2 | 25.0 | 6th (—) |
| J2-Jumbo-Instruct (178B*) | 32.6 | 33.7 | 19.2 | 12th (—) | 27.3 | 23.5 | 31.2 | 37.1 | 32.0 | 32.7 | 51.3 | 25.0 | 7th (—) |
| ChatGLM (130B) | 38.0 | 56.5 | 36.1 | 5th (↑2) | 30.5 | 47.8 | 51.9 | 16.0 | 25.0 | 23.3 | 30.3 | 25.0 | 8th (—) |
| FLAN-T5 (11B) | 56.1 | 51.5 | 32.9 | 4th (↓1) | 63.2 | 47.8 | 49.1 | 18.7 | — | — | — | 25.0 | 5th (—) |
| InstructGPT curie v1 (6.7B*) | 28.1 | 43.8 | 28.9 | 10th (↓1) | 29.4 | 41.2 | 41.8 | 22.3 | 25.4 | 22.0 | 25.8 | 25.0 | 9th (—) |
| LLaMa (65B) | 24.2 | 25.6 | 19.5 | 15th (↓1) | 22.0 | 18.4 | 18.3 | 55.6 | 31.4 | 30.1 | 25.5 | 25.0 | 10th (↑1) |
| ChatGLM2-32k (6B) | 25.3 | 22.8 | 20.1 | 16th (—) | 22.0 | 40.8 | 20.0 | 19.0 | 26.5 | 20.6 | 22.7 | 25.4 | 17th (—) |
| Alpaca (7B) | 21.5 | 25.3 | 18.2 | 17th (↓2) | 22.0 | 18.4 | 18.8 | 25.3 | 26.4 | 31.4 | 22.2 | 25.0 | 20th (—) |
| Llama2-chat (7B) | 21.6 | 19.7 | 17.9 | 22th (↓3) | 25.2 | 18.4 | 24.5 | 37.4 | 32.5 | 20.6 | 27.2 | 25.6 | 14th (—) |
| ChatGLM (6B) | 31.9 | 32.9 | 30.5 | 11th (—) | 23.1 | 45.5 | 32.9 | 16.0 | 25.0 | 22.0 | 22.8 | 25.0 | 13th (—) |
| Vicuna (13B) | 18.8 | 19.1 | 17.4 | 26th (—) | 22.0 | 18.7 | 23.3 | 29.8 | 26.0 | 35.4 | 30.5 | 25.0 | 15th (—) |
| GLM (130B) | 21.9 | 25.1 | 22.9 | 14th (↑3) | 22.0 | 18.4 | 18.3 | 49.6 | 33.2 | 29.7 | 22.2 | — | 12th (↓2) |
| GPT-J (6B) | 20.8 | 18.8 | 18.0 | 25th (↓1) | 22.0 | 18.4 | 18.3 | 22.2 | 25.0 | 31.0 | — | 25.0 | 25th (—) |
| T0++ (11B) | 41.9 | 38.4 | 23.9 | 9th (↑1) | 30.5 | 39.2 | 27.8 | 16.0 | — | — | — | 25.0 | 16th (—) |
| Dolly-v2 (12B) | 21.1 | 21.0 | 18.9 | 20th (↑2) | 22.0 | 18.4 | 18.8 | 28.5 | 25.0 | 20.6 | 27.1 | 25.0 | 23th (—) |
| GPT-JT (6B) | 19.8 | 18.9 | 19.0 | 24th (↑1) | 22.0 | 18.4 | 18.3 | 19.2 | 25.0 | 36.7 | — | 25.0 | 22th (—) |
| Internlm-chat-8k (7B) | 23.5 | 20.4 | 17.2 | 19th (↑1) | 22.0 | 18.4 | 18.3 | 19.0 | 27.2 | 20.6 | 26.1 | 25.0 | 27th (—) |
| UL2 (20B) | 26.5 | 28.1 | 18.9 | 13th (—) | 22.0 | 18.4 | 18.3 | 18.0 | — | — | — | 25.0 | 28th (—) |
| GPT-3 davinci v1 (175B) | 18.1 | 17.9 | 16.9 | 27th (—) | 22.0 | 18.7 | 18.3 | 30.2 | 25.0 | 29.6 | 22.3 | 25.0 | 19th (—) |
| GPT-NeoX (20B) | 19.9 | 20.7 | 18.2 | 23th (—) | 22.0 | 18.4 | 18.3 | 25.7 | 25.0 | 32.1 | — | 25.0 | 21th (—) |
| BLOOM (7B) | 21.0 | 21.9 | 18.3 | 18th (—) | 22.0 | 18.4 | 18.3 | 30.1 | 28.0 | 29.2 | 22.2 | 25.0 | 18th (—) |
| GPT-3 curie v1 (6.7B) | 17.2 | 17.7 | 16.8 | 28th (—) | 22.0 | 18.4 | 18.3 | 21.5 | 25.0 | 25.6 | 23.9 | 25.0 | 26th (—) |
| RedPajama-Instruct (7B) | 21.8 | 21.2 | 16.4 | 21th (—) | 22.0 | 18.4 | 18.3 | 29.8 | 25.0 | 20.6 | 25.4 | | 24th (—) |

Table 3: Standardized performance of Knowledge Applying, Creating level and all the 4 levels.

| Model | Level 3: KA | | | | | | | Level 4: KC | | | Overall (1,2,3,4) | |
|---|---|---|---|---|---|---|---|---|---|---|---|---|
| | 3-1 | 3-2 | 3-3 | 3-4 | 3-5 | 3-6 | Rank | 4-1 | 4-2 | Rank | Avg | Rank |
| GPT-4 | 59.8 | 60.7 | 76.8 | 32.8 | 60.9 | 58.1 | 1st (—) | 48.6 | 58.9 | 2nd (↑1) | 2.33 | 1st (—) |
| GPT-3.5-turbo | 58.5 | 42.6 | 53.9 | 45.6 | 30.9 | 19.6 | 5th (↓1) | 52.2 | 46.4 | 3rd (↓1) | 1.34 | 2nd (—) |
| InstructGPT davinci v2 (175B*) | 30.6 | 39.7 | 44.2 | 23.6 | 50.3 | 23.4 | 7th (↓1) | 54.4 | 56.8 | 1st (—) | 1.12 | 3rd (—) |
| Tulu (7B) | 42.5 | 45.6 | 40.7 | 54.8 | 42.3 | 54.8 | 3rd (↑2) | 32.8 | 42.5 | 9th (↑2) | 0.64 | 4th (↑3) |
| Cohere-command (52.4B) | 36.2 | 41.7 | 45.3 | 49.3 | 54.7 | 44.0 | 4th (↓1) | 17.1 | 15.1 | 25th (↓13) | 0.54 | 5th (↓1) |
| FLAN-UL2 (20B) | 49.6 | 47.5 | 39.4 | 51.1 | 43.5 | 53.3 | 2nd (—) | 28.3 | 19.0 | 20th (↓3) | 0.54 | 6th (↓1) |
| J2-Jumbo-Instruct (178B*) | 45.2 | 31.6 | 31.6 | 38.3 | 28.3 | 25.5 | 8th (—) | 43.7 | 53.3 | 4th (—) | 0.47 | 7th (↑1) |
| ChatGLM (130B) | 36.4 | 34.1 | 28.0 | 36.4 | 36.7 | 21.2 | 9th (↑1) | 24.4 | 28.4 | 16th (↑2) | 0.28 | 8th (↑1) |
| FLAN-T5 (11B) | 45.0 | 49.0 | 32.9 | 51.1 | 39.7 | 16.1 | 6th (↑1) | 20.2 | 0.0 | 28th (↓6) | 0.22 | 9th (↓3) |
| InstructGPT curie v1 (6.7B*) | 31.6 | 37.2 | 24.3 | 29.1 | 31.2 | 27.2 | 11th (—) | 27.7 | 29.6 | 12th (↑3) | 0.06 | 10th (—) |
| LLaMa (65B) | 16.4 | 35.4 | 41.7 | 25.4 | 21.7 | 17.6 | 16th (↓2) | 44.7 | 37.1 | 5th (—) | 0.01 | 11th (—) |
| ChatGLM2-32k (6B) | 35.7 | 31.1 | 24.0 | 40.1 | 20.7 | 17.3 | 13th (↓1) | 34.5 | 41.5 | 8th (—) | -0.09 | 12th (↑2) |
| Alpaca (7B) | 15.1 | 19.3 | 20.9 | 18.1 | 45.5 | 50.7 | 12th (↑5) | 35.7 | 41.0 | 7th (↑3) | -0.12 | 13th (↑2) |
| Llama2-chat (7B) | 17.5 | 16.8 | 23.4 | 14.4 | 14.4 | 51.7 | 14th (↑2) | 31.7 | 38.1 | 10th (↓4) | -0.18 | 14th (↓2) |
| ChatGLM (6B) | 21.2 | 27.5 | 22.2 | 19.9 | 19.5 | 28.5 | 20th (—) | 17.7 | 30.8 | 18th (↑6) | -0.24 | 15th (↑5) |
| Vicuna (13B) | 25.4 | 10.0 | 24.7 | 18.1 | 21.0 | 16.4 | 24th (↓1) | 35.0 | 45.9 | 6th (↑1) | -0.26 | 16th (↑1) |
| GLM (130B) | 23.5 | 13.0 | 18.4 | 21.7 | 45.0 | 35.1 | 17th (↓4) | 29.3 | 19.1 | 19th (↓3) | -0.33 | 17th (↓1) |
| GPT-J (6B) | 38.8 | 39.4 | 26.8 | 49.3 | 17.5 | 16.5 | 10th (↓1) | 30.5 | 24.0 | 14th (↑9) | -0.33 | 18th (↑3) |
| T0++ (11B) | 22.4 | 23.0 | 23.8 | 14.4 | 39.7 | 16.1 | 19th (—) | 18.1 | 3.7 | 27th (↓8) | -0.44 | 19th (↓6) |
| Dolly-v2 (12B) | 14.1 | 20.5 | 18.4 | 18.1 | 26.4 | 25.2 | 22th (—) | 29.5 | 31.9 | 11th (↓2) | -0.45 | 20th (↓1) |
| GPT-JT (6B) | 31.4 | 38.2 | 23.0 | 32.8 | 18.5 | 17.5 | 15th (—) | 21.2 | 19.5 | 21th (↑5) | -0.54 | 21th (↑3) |
| Internlm-chat-8k (7B) | 19.3 | 20.8 | 22.8 | 14.4 | 17.1 | 22.2 | 23th (↑1) | 26.1 | 27.8 | 15th (↑5) | -0.56 | 22th (↑1) |
| UL2 (20B) | 25.2 | 28.0 | 25.9 | 38.3 | 16.1 | 16.1 | 18th (—) | 26.9 | 9.0 | 23th (↓9) | -0.56 | 23th (↓5) |
| GPT-3 davinci v1 (175B) | 18.3 | 13.5 | 22.8 | 19.9 | 21.5 | 17.0 | 25th (—) | 32.3 | 22.6 | 13th (—) | -0.57 | 24th (↓2) |
| GPT-NeoX (20B) | 14.0 | 13.9 | 17.1 | 18.1 | 22.7 | 16.7 | 28th (↓1) | 32.2 | 19.2 | 17th (↑4) | -0.61 | 25th (—) |
| BLOOM (7B) | 18.6 | 22.6 | 17.1 | 19.9 | 27.4 | 23.3 | 21th (—) | 17.9 | 21.4 | 22th (↑5) | -0.61 | 26th (—) |
| GPT-3 curie v1 (6.7B) | 22.6 | 15.2 | 20.5 | 18.1 | 19.1 | 16.4 | 26th (—) | 23.7 | 10.1 | 24th (↑1) | -0.81 | 27th (—) |
| RedPajama-Instruct (7B) | 12.6 | 10.0 | 17.1 | 14.4 | 26.1 | 23.2 | 27th (↑1) | 13.7 | 12.3 | 26th (↑2) | -0.85 | 28th (—) |

and the model size. This suggests that model size has an obvious positive impact on memorizing seen knowledge, which corroborates some of the viewpoints from previous studies (Liang et al., 2022).

(2) For models after instruction tuning, there is a significant increase in the correlations between higher-level abilities and model size (exemplified by KA, whose Spearman's coefficient 0.02 to 0.53). This suggests that alignment unleashes the greater potential of LLMs in higher-level capabilities. However, the correlation between size and low-level KM performance exhibits a decline (0.34), potentially demonstrating the widely discussed "alignment tax" (Ouyang et al., 2022).

(3) Compared to the commercial closed-source models like GPT4 and GPT-3.5-turbo, there is still a noticeable gap in the performance of open-source models. Open-source models obtain an average z-score of $-0.29$, which is below the overall average. Comparing the second-season results with the first season, the rankings of most open-source models have declined. This suggests that static open-source models struggle to maintain a comparable level with potentially continuously updated commercial models in the long run. The open-source community should advocate for stronger collaborations to support larger and up-to-date models that are crucial for future research purposes.

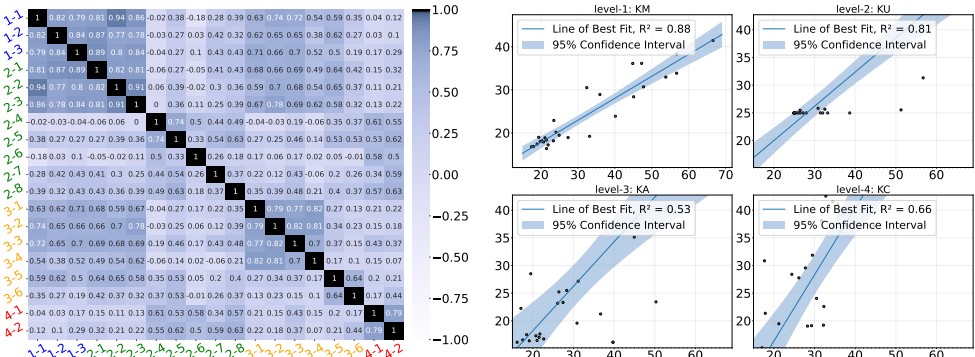

Figure 3: **Left**: The *spearman* correlation coefficient. Each cell represents the correlation of model rankings on two tasks. **Right**: Scatter plots of rolling task vs. corresponding non-rolling task (e.g., 3-5 v.s. 3-6). The x-axis and y-axis of each subplot represent the standard scores correspondingly.

**Design Analysis.** We further discuss several new observations brought by KoLA design factors.

First, there is a high correlation among tasks within each level, indicating that the abilities of LLM indeed possess some inherent hierarchical structure. The knowledge memorization (KM) level shows notable correlations with other levels, especially with the concept tasks in the understanding level (2-1, 2-2, 2-3), as well as with the reasoning tasks (from 3-1 to 3-5) in the applying level, which indicates that these high-level tasks rely heavily on knowledge memory. Moreover, in order to obtain a more dissociated assessment of the LLMs' competence in higher-order cognitive tasks, it is still recommended to design tasks that exhibit substantial disparities from the pre-training corpus to alleviate the potential biases stemming from data.

Second, the results of the models on evolving and non-evolving tasks show an obvious linear correlation, indicating the reliability of our construction of evolving datasets. The performance gap between known and evolving data is more prominent for shallower levels (KM, KU), whereas it is less pronounced in higher-level tasks (KA, KC). The convergence of performance between Independent-Identical-Distribution and Out-of-Distribution evolving settings suggests a potential enhancement in the model's generalization capability and may support the opinion about the model's acquisition of divergent and reasoning abilities that go beyond simple data fitting (Bubeck et al., 2023; Zhong et al., 2023).

Third, we conduct manual annotation (Appendix F.1 for more details about annotation settings and results) on the results in the knowledge creating tasks, where each annotator is required to read the contexts $C$ and foreknowledge $K$, and then evaluate the model's outputs $T$ in two aspects: overall quality and faithfulness. Ratings are assigned on a scale of $1$ (the worst rating) to $5$ (the best rating). We calculate Spearman's correlation between the manually annotated results and the metrics introduced in § 2.3. We find that there is a notable correlation ($0.61$) observed between the self-contrast metric $\partial(T, T_k)$ and the faithfulness of created content, while removing the self-contrast metric from the overall metric $x$ in Eq. (3) brings a significant 32% decrease in the correlation with the human-judged overall quality. We believe this metric can contribute to future explorations on the assessment of generation abilities (Theis et al., 2016; Pillutla et al., 2021; Fu et al., 2023).

## 4 CONCLUSION AND FUTURE WORK

This paper presents KoLA, a carefully designed Knowledge-oriented LLM assessment benchmark. We design a cognitive ability taxonomy for more helpful diagnostic results, adopt both known and evolving data sources for better fairness, and employ contrastive metrics for high applicability. In the first season of KoLA, we evaluate $28$ open and commercial LLMs and get some intriguing findings, such as larger models tend to memorize more knowledge, and alignment unleashes the potential of higher-level abilities but may harm the low-level knowledge memorization, etc. In the future, we will continually host more seasons of KoLA to facilitate knowledgeable LLMs, help select backbones for developing knowledge-related applications, and track the development of LLMs with evolving evaluations. KoLA will always welcome open participation and contributions.

## ACKNOWLEDGEMENT

This research project is supported by a grant from the Institute for Guo Qiang, Tsinghua University (2019GQB0003). Jie Tang is supported by Technology and Innovation Major Project of the Ministry of Science and Technology of China under Grant 2022ZD0118600, NSFC for Distinguished Young Scholar 61825602 and the New Cornerstone Science Foundation through the XPLORER PRIZE.

This project also appreciate the providing of the test set of MusiQue (Trivedi et al., 2022) by Harsh Trivedi, Stony Brook University and the test set of 2Multiwikihop (Ho et al., 2020) by Xanh Ho, National Institute of Informatics, Tokyo, Japan.

## ETHICS STATEMENT

In this section, we discuss the ethical considerations regarding our data construction and leave the broader impact to Appendix A.2. (1) **Data Risk Control.** Regarding the collected evolving data source, we have filtered out content that is inappropriate for presentation to a general audience, and the relevant details are outlined in Appendix C.1. Seven of the authors manual check all the newly constructed evolving test datasets as well as random samples of all the previously released datasets included in KoLA. No instances of personally identifiable information, discriminatory content, explicit, violence, or offensive content were found. (2) **Annotator Treatment and Consent.** We hire crowdsourced annotators in the annotation of evolving test data and the human evaluation for knowledge creating. The details are introduced in Appendix C.2. We have signed work contracts with all the annotators and provided compensation according to mutually agreed-upon wage standards and working hours. All employment arrangements are in compliance with local regulations. (3) **Copyright.** Our known data source is Wikipedia, which is licensed under CC BY-SA 3.0[8] and allows for free research use. For all the previously released datasets included in KoLA. Our evolving data source contains public news and fictions. The news data is from The Guardian[9] and we access it strictly following the terms and conditions[10]. The fiction data is from Archive of Our Own (AO3)[11], a fan-fiction archive site. Although AO3 data has been used in some previous works Cao & Daumé (2020); Yoder et al. (2021); Sun et al. (2022); Krishna et al. (2022), there remains some ambiguity regarding its copyright status. We believe our use of AO3 is appropriate because: (i) AO3 exhibits an open attitude towards data crawling[12]. (ii) We pledge that KoLA will always remain non-commercial and non-profit, and we do not redistribute the crawled data (only samples are provided in our platform). According to the description[13] provided by the Organization for Transformative Works, the operator of AO3, such usage falls under fair use in the context of the U.S. copyright law.

## REPRODUCIBILITY STATEMENT

To promote reproducibility, we provide details about our data collection in Appendix C, all the used task instructions in Appendix D, and experimental details in Appendix E. The evaluation source codes and data samples for all the tasks are submitted as supplementary material. The results of future seasons will be presented at the Github and our platform website.

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

APPENDICES

# A   BROADER DISCUSSION

## A.1   LIMITATION

The major limitation of KoLA is that our coverage is not as extensive as some other recent works (Hendrycks et al., 2021; Zhong et al., 2023; Huang et al., 2023). KoLA evaluates LLMs' world knowledge about concepts, entities, and events and only covers 19 English datasets now. While there is no doubt that expanding the evaluation "breadth" to test the boundaries of LLM abilities is valuable, our emphasis lies more on the "depth" of evaluation. Due to careful design considerations regarding data sources and our annotation capacity to host a new competition season every 90 days, it is not easy to significantly broaden our evaluation coverage. The first season results reported in this paper involve 21 LLMs. Although we strive to cover diverse and representative LLMs, it is challenging to cover the ever-emerging and evolving LLMs with solely our own efforts. We sincerely welcome community contributions and participations to introduce new tasks or LLMs.

Another limitation pertains to our evaluation of knowledge creating abilities. To avoid human evaluations, as detailed in Section 2.2 and Section 2.3, we design an automatic evaluation method based on the self-contrast metric. Experiments in Section 3 validate the efficacy of our metric. However, our automatic evaluation still relies on contrasting model-generated content with human-provided knowledge. If a model produces knowledge that is novel and reasonable but just not aligned with human-provided knowledge, its capabilities might be underestimated. We encourage future work to actively explore this significant avenue and endeavor to develop more effective automatic evaluation methods.

## A.2   POTENTIAL IMPACT

As a benchmark, the intended use of KoLA is not to construct applications or train LLMs, but rather to evaluate the foundational abilities about world knowledge of LLMs. Our evaluation tasks do not involve speculating personal sensitive information, making judgments on social issues, or interacting with the real world. Therefore, we believe the likelihood of our benchmark directly leading to negative impacts on safety, security, discrimination, surveillance, deception & harassment, human rights, bias and fairness is very low. However, effective benchmarks will facilitate the development of powerful LLMs, which poses a wide and serious risk of misuse. Although beyond the scope of this paper, we earnestly call for strengthened cooperation from various sectors of society in enhancing the regulation and safety control of LLMs.

Our evaluation needs to do inference with many LLMs on various datasets, which naturally results in carbon emissions and potential environmental issues. The total carbon emissions can be estimated based on the data provided in Appendix E.1. As our evaluation does not involve the pre-training and fine-tuning of LLMs, we believe the impact caused is relatively marginal and controllable. In the participation guidelines on the platform, we also state that we discourage improving evaluation scores through repeated submissions or training specifically on benchmark-related data. This ensures the reliability of the evaluation results while minimizing carbon emissions as much as possible.

## B    AUTHOR CONTRIBUTION

**Data Collection.** Xiaozhi Wang, Shulin Cao, Xin Lv, Hao Peng, Zijun Yao collected the *Known Data Source* from the open-source projects, private academic datasets and Wikidata. Jifan Yu, Daniel Zhang-Li, Nianyi Lin, Linlu Gong and Kaifeng Yun collected and pre-processed the first season's *Evovling Data Source*. Hailong Jin supported the data from Xlore (Jin et al., 2019). Yuan Yao provided the test set of DocRED (Yao et al., 2019) and Ning Ding helped the construction of FewNERD (Ding et al., 2021) test set.

**Data Annotation.** Xiaozhi Wang organized the crowdsourcing annotation of fine-grained event arguments for knowledge creating (KC) tasks. Kaisheng Zeng and Yong Guan assisted in the quality control process. Jifan Yu and Daniel Zhang-Li organized the annotation of knowledge triples from the evolving articles, as well as the human evaluation of creating results.

**Task Construction.** Xin Lv designed the dataset and instruction of the knowledge memorization (KM) tasks, (1-1) and (1-2). Hao Peng organized the knowledge understanding (KU) tasks and constructed instructions for (2-1/2-2/2-3). Zhili Wu, Yunjia Qi, Jianhui Chen and Weikai Li correspondingly constructed instructions for task (2-4) to (2-7). Shulin Cao and Zijun Yao organized the knowledge applying (KA) tasks and constructed instructions for (3-1), (3-3), (3-4) and (3-5). Yantao Liu and Amy Xin constructed task (3-2) and (3-6). Nianyi Lin and Daniel Zhang-Li organized the knowledge creating (KC) tasks (4-1), (4-2) and most of the evolving tasks. Kaifeng Yun and Linlu Gong constructed instructions for task (1-3) and (2-8).

**Model Evaluation.** Shangqing Tu organized the whole process of model evaluation. Hanming Li deployed and conduct experiments of GPT-J (6B), GPT-JT (6B), GPT-NeoX (20B), BLOOM (7B), T0++ (11B), LLaMa (65B), GLM (130B), UL2 (20B), FLAN-T5 (11B), Alpaca (7B), FLAN-UL2 (20B), ChatGLM (6B). Chunyang Li evaluated GPT-3 curie v1 (6.7B) and davinci v1 (175B), InstructGPT curie v1 (6.7B*) and davinci v2 (175B*), ChatGLM (130B), and GPT3.5-turbo. Zheyuan Zhang evaluated Cohere-command (52.4B) and J2-Jumbo-Instrcut (178B*), while Yushi Bai evaluated GPT-4. Ji Qi, Daniel Zhang-Li and Jinxin Liu assisted the data analysis and presentation on the inference results. Yu Gu supported some of the candidate APIs.

**Platform Development.** Xiaohan Zhang's team organized the overall platform development. Jifan Yu and Daniel Zhang-Li respectively contributed to the visualization design and backend development.

**Paper Writing.** Jifan Yu and Xiaozhi Wang hosted the paper writting. Shangqing Tu, Ji Qi, Daniel Zhang-Li supported the part of experimental analysis. All the authors contributed to their corresponding working details for completing this paper (including Appendix).

**Advising.** Lei Hou, Zhiyuan Liu, Bin Xu, Jie Tang and Juanzi Li take advisor roles in this project. Juanzi Li is the main advisor, initializing, supporting and organizing this project.

This project also express appreciations to other supporters. Special thanks go to Gang Wang for his outstanding product prototype design. The artistic design, including the beautiful illustrations for the paper, was skillfully provided by Shanshan Wang. The platform development and feature implementation were carried out by Zhenfang Lu, Shuai Xie, Shuaiming Wang, Liangliang Cui and Dingxiao Liu. The coordination of crowdsourcing affairs was effectively managed by Jupeng Zhang and Yue Yang. Finally, the long-term assistance and support from Yini Chen to the AIGC research team are greatly acknowledged.

## C    DETAILS OF DATA COLLECTION

There are two data sources used in KoLA: *Known* and *Evolving*, which are correspondingly from Wikipedia and newly crawled corpus. In this section, erview of the collection and maintenance of them, as well as the annotation process involved, including essential statistical information.

### C.1    RAW DATA COLLECTION

**Known**. We collect the corresponding Wikipedia articles for the entities in Wikidata5M (Wang et al., 2021a) from Xlore2 (Jin et al., 2019), a cross-lingual knowledge base in Chinese and English, using its 2019 version, and align them accordingly. This process generates a dataset of 5 million

articles. Given that language model training is reliant on textual data, we depart from the conventional graph-based methods typically employed in knowledge graphs to calculate entity frequencies. Instead, we conduct statistical analysis to determine and rank the occurrence frequencies of these entities and their aliases within the text corpus. Subsequently, we establish two sets of high-frequency and low-frequency entities, each containing 2,000 entities, to fulfill the requirements of the knowledge memorization task 1-1 and 1-2.

**Evolving**. The data collection process for the Evolving dataset in the first season of KoLA concludes on April 15, 2023. Therefore, we are collecting data from the preceding 90 days (January 15, 2023, to April 14, 2023). In terms of news data, we are experimenting with multiple open-source news scraping interfaces. Our primary focus is on gathering articles that have rich event elements, such as factual news and entertainment news. We have collected a total of 1,000 such articles, aiming to avoid sensitive news categories like politics. As for the novel data, we randomly selected 1,000 works from a renowned creative platform[14], striving to achieve a balanced representation of various writing types (fan fiction and original works). Afterward, we employ an open-source tool[15] to filter out chapters containing explicit, violent, or other inappropriate content, sorting them based on the number of views. Following this filtering process, we ultimately retain 250 articles each from the news and novel categories as candidates (500 in total) for high-quality texts.

## C.2   DATA ANNOTATION

During the construction of the KoLA benchmark, we have two key annotation tasks: (1) Fact Triple Annotation and (2) Event Argument Annotation. The annotated triples from (1) will subsequently be utilized for the construction of the evolving test sets for Task 1-3, 2-8, and 3-6. Meanwhile, the annotated event attributes from (2) will be employed for Task 4-1 and 4-2. Throughout the annotation process, we meticulously provided comprehensive instruction documents and implemented reasonable model pre-annotations, aiming to direct the annotators' focus to the most essential parts, thereby striving to enhance the quality of the primary annotated data. Subsequent to this, we also conducted multiple rounds of quality checks to ensure the reliability of the final constructed test set.

**Annotation Team**. We hire a 21-member annotation team (based on market rates) comprising experienced annotators. With their permission, we gather some basic information about the annotation team and present it in Table 4. In general, the annotation team consists mostly of individuals with graduate-level qualifications. The validators are three Ph.D. holders from the KoLA team. With the collaboration of this high-quality team, we strive to ensure the efficiency and quality of data annotation. Before initiating the annotation work, we enter into a legally binding contract with the team to protect the rights of the annotators. We also develop a dedicated platform specifically for the annotation process, which facilitates efficient review, publishing, and exporting of the annotation results. The platform allows annotators to have flexibility in choosing their working hours, including the ability to save intermediate results, retrieve relevant resources, and log in or log out at any time.

Table 4: Statistics of the annotation team for dataset construction in KoLA.

| Gender | Rate |
| --- | --- |
| Female | 85.7% |
| Male | 14.3% |

| Education | Rate |
| --- | --- |
| Bachelor | 47.6% |
| Master | 52.4% |

**Annotation Process and Quality Control**. For the aforementioned annotation tasks, we carefully employ methods for data pre-processing to facilitate the annotation process and quality control.

For Task (1), the text to be annotated includes only the Evolving data. We take the following steps to obtain the factual knowledge triples and whether they can be infered from the previous corpus.

1. **Named Entity Recognition**. We first utilize a named entity recognition tool[16] to extract entities from the articles. This model exhibits strong performance on four entity types (PER, ORG, LOC, MISC) with a Precision of 90.7%, Recall of 91.9%, and an F1-score of 91.3%.

---

[14] https://archiveofourown.org

[15] https://huggingface.co/unitary/toxic-bert

[16] https://huggingface.co/dslim/bert-base-NER

After removing a small percentage of incompletely recognized ones ($< 5.3\%$), these results are used as input for subsequent processes.

2. **Relation Extraction**. Subsequently, we replicate a renowned document-level relation extraction model, ATLOP (Zhou et al., 2021), using these entities and documents to extract potential relationships and organized them into triple formats. Due to the task's complexity, the model achieves only an F1 score of $63.4\%$. Therefore, during replication, we reduce its prediction threshold to $0.4$ further enhance the model's Recall (to $84.5\%$), aiming to minimize the omission of triples during the annotation process.

3. **Triple Annotation**. After the aforementioned preprocessing, a total of 23.1k candidate triples entered the annotation phase. Each annotator is permitted to use online searches and is instructed to categorize each candidate triple into the following three classes: a) incorrect triples, b) correct triples that can be known prior to the current season, and c) correct triples that can only be known after the current season began. Each triple is guaranteed to be annotated by two annotators. Eventually, the Cohen's Kappa for the correctness of the triple annotations (a vs b+c) is $0.71$, and the Cohen's Kappa for whether the triple could be discovered before January 15th (b vs c) is $0.55$, indicating a fair good agreement on such issues. In cases of classification discrepancies, the decision is deferred to the quality check lead.

For Task (2), the text to be annotated includes both Known and Evolving data. As for the Evolving data, we also conduct processing as:

1. **Event Detection**. We also employ the named entity recognition results (as in the annotation task (1)) to reproduce the Omni-Event toolkit[17]. Specifically, we replace the backbone model with CLEVE (Wang et al., 2021b) for event detection, which achieves a Recall of $81.5\%$ on the ACE2005 dataset and $72.6\%$ on MAVEN dataset. The detected events are subsequently used as candidates for event argument annotation.

The Known data portion utilizes articles from the MAVEN dataset (Wang et al., 2020), where the event trigger words and event types have already been annotated, requiring no additional pre-processing. Then, articles with detected events from both known and evolving data are processed into annotations.

2. **Event Argument Annotation**. Annotators are required to perform the following tasks: a) annotate candidate attributes for each event, and b) correct or delete events if the event trigger words are incorrect. To guarantee the annotation quality, we build an online annotation platform that indexes examples for all $159$ event types (with an average of 7 argument roles per type). The annotated results subsequently underwent two rounds of quality checks. Annotations deemed unsatisfactory in each round are returned for re-annotation. The first-time pass rate for the first round of quality checks is $67\%$, and $91\%$ for the second round. The overall Fleiss' Kappa for the annotation is $0.62$. Ultimately, on average, $57.8$ events and $235.3$ arguments were annotated per article.

## C.3 AVAILABILITY

**Platform**. We develop an online platform to offer a range of services to the community, such as competition news updates, visualizations of evaluation results, and convenient access to submit new models or modify previous submissions. Due to the dynamic nature of the KoLA evolving data source, new results and rankings will be generated in each season, and a selection of results from previous seasons will also be publicly available.

**Participate in KoLA.** Researchers can participate in KoLA evaluations in two roles. (1) *Competitor*: KoLA welcomes open participation for each season by providing the model's APIs or parameters, and provides 5 high-quality examples for each task to help participants debug. It is worth noting that KoLA does not allow the local evaluation to prevent test set leakage and unfairly overfitting datasets. (2) *Contributor*: We maintain a special interest group (invitation link shown at the "About" page of the platform) where volunteers who have ideas for result analysis, model refinement, and benchmark improvement can discuss, propose suggestions, and participate in task construction.

---

[17]https://github.com/THU-KEG/OmniEvent

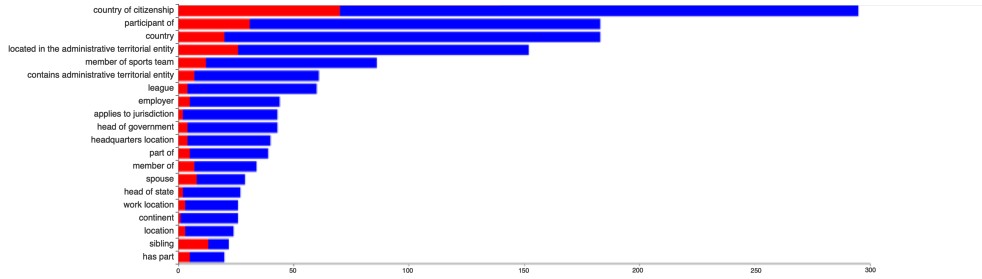

Figure 4: The distribution of the top-20 annotated triples in terms of relation type. Blue bars represent the correct triples that can be known prior to the current season, and red bars represent the correct triples that can only be known after the current season begins.

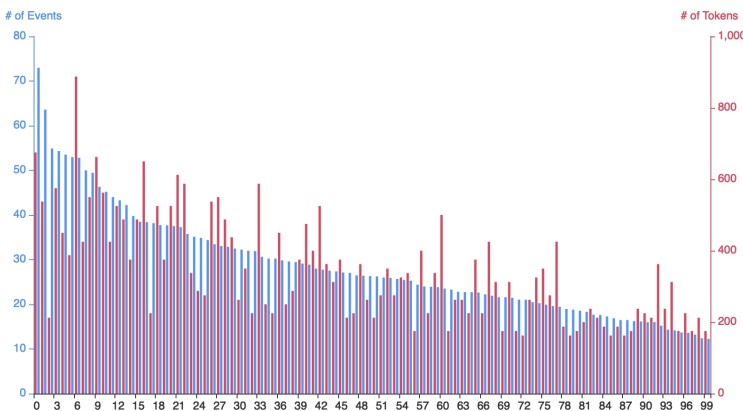

Figure 5: The distribution of the events and text length in the articles from Wikipedia (MAVEN). Blue bars and red bars correspond to the number of events and the tokens in each article.

**Supporting Tools.** We release a toolkit to support KoLA-related functions at Github, including: (1) *Easy-to-submit*. Competitors can employ this function to independently maintain the in-context prompts for each task while providing a single model API, making the submission and modification convenient. (2) *Result Reproduction*. We provide the code and developed tools that used in our data visualization and standardization, which support result reproduction and other analyses. (3) *Data Acquisition*. We also provide a data access API, which assists authorized users in getting the evolving data and results of previous seasons.

### C.4    DISTRIBUTION OF THE ANNOTATED RESULTS

For Task (1), we annotate all 500 articles and retained 2.7K correct triples, out of which only 459 triples cannot be found in earlier corpora. Figure 4 illustrates the distribution of relations for all correct triplets, where blue bars represent category b) correct but can be known before the certain season; and red bars are the number of category c) correct and cannot be known before the certain season. It can be observed that even in the Evolving data, the triplets still exhibit a long-tail distribution. Most of the triplets, such as "country of citizen" and "country", do not effectively convey the main content of the articles. This further reinforces our goal of annotating fine-grained event-level knowledge.

For Task (2), we specifically select 100 articles from the MAVEN and Evolving datasets, ensuring they possess extensive knowledge (i.e., a substantial number of entities and event triggers that are not within the lowest 20% frequency range). After completing the annotation process, we examine the number of valid events (events containing at least one argument) and the article lengths, as illustrated in Figure 5 and Figure 6.

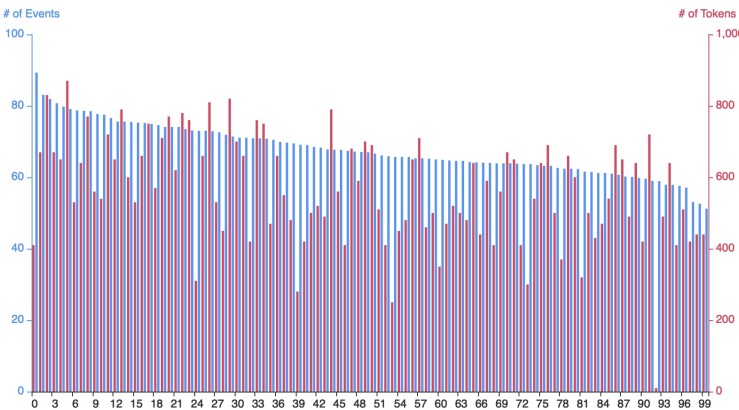

Figure 6: The distribution of the events and text length in the articles from *Evolving Data*.

In general, the distribution of event knowledge between the two datasets is quite similar. The articles in the Evolving dataset are generally longer compared to the Wikipedia articles in MAVEN, resulting in a higher number of valid events. Due to this difference, models may encounter more challenges when performing tasks on the Evolving dataset, but they may also benefit from greater exposure to knowledge. Therefore, the varying performance of models on the tasks upon these two data sources may be influenced by factors such as the model's parameter size or training adequacy. This phenomenon warrants further analysis and discussion.

## C.5 ANNOTATION COST

To sustain the project over the long term, we have kept the budget for each season below USD 2,000. Generally, each season's expenditure comprises two main parts: 1) the cost of data annotation, and 2) the expenses for model deployment and API calls.

The data annotation includes labeling knowledge triplets and event arguments. As introduced in Appendix C.2, to reduce the difficulty and cost of annotation, we have pre-labeled the collected text automatically, allowing annotators to focus on the core tasks. Overall, each season requires approximately USD 660-700 for triple annotation and USD 400-420 for event annotation.

Due to the scale of the KoLA test sets, the costs for model deployment and API calls have been kept within an acceptable range. The deployment expenses, estimated from GPU usage time, are approximately USD 200-240, while the API incurs an additional cost of USD 150-180.

Overall, for the two seasons completed thus far, the costs have not exceeded USD 1,600 per season. With the anticipated increase in the number of models in future seasons, we believe that a budget of USD 2,000 per season should be sufficient to maintain the project.

# D DETAILS OF TASK INSTRUCTION

After completing the data collection, we proceed to construct separate test sets for tasks at different levels, thereby transforming knowledge-related tasks into language tasks driven by instructions, which facilitates the execution by large-scale models. In this section, we first present the design principles for each level of tasks, followed by specific approaches to constructing detailed instructions, and provide corresponding task examples.

## D.1 CONVERTING TO TASK FORMAT

There are 7 tasks' test sets that need to be constructed from scratch ((1-1) High-Freq., (1-2) Low-Freq., (4-1) Encyclopedic Knowledge Creating and 4 Evolving Test tasks). These tasks require the reconstruction and quality control of the data based on our annotations. As for the other 12 tasks, we only focus on designing how they can be transformed into sequence tasks that can be solved by language models, considering the dataset construction methods provided in the original text as references. Overall, we follow two principles during the process of constructing instructions: a) Simplicity: We aim to describe the task objectives using the least amount of text, thus saving the model's in-context length; b) Standardization: We use special markers to identify all structured knowledge, assisting the model in quickly capturing the knowledge objectives.

## D.2 KNOWLEDGE MEMORIZATION TASKS

Knowledge Memorization (KM) level primarily assesses the model's ability to retain knowledge triples. However, this format is not inherently suitable for large models. Therefore, we transform the triple prediction task into a question-answering task that considers 1-to-N relationships. For each type of relationship, we design specific templates to facilitate this transformation.

| |
|---|
| **INSTRUCTION:** Please give answers to the following questions about knowledge. Note: If there are more than one answer, print them all and separate them with a semicolon (;). Please do not give anything other than the answers. |
| **QUESTION:** What is the occupation of Wang Guozhen? |
| **ANSWER:** poet |

Table 5: The instruction and an example of Task 1-1 High-Freq. KM.

| |
|---|
| **INSTRUCTION:** Please give answers to the following questions about knowledge. Note: If there is more than one answer, print them all and separate them with a semicolon (;). Please do not give anything other than the answers. |
| **QUESTION:** Which country does White Hall Township belong to? |
| **ANSWER:** United States of America |

Table 6: The instruction and an example of Task 1-2 Low-Freq. KM.

| |
|---|
| **INSTRUCTION:** Please give answers to the following questions about knowledge. Note: If there is more than one answer, print them all and separate them with a semicolon (;). Please do not give anything other than the answers. |
| **QUESTION:** Will Messi still serve in Paris Saint-Germain? |
| **ANSWER:** No |

Table 7: The instruction and an example of Task 1-3 ETM KM.

## D.3 KNOWLEDGE UNDERSTANDING TASKS

Knowledge Memorization (KM) level involves various levels of structured information, such as concepts, entities, relationships, and events. It also incorporates multiple documents, even at the multi-document level, which can easily overwhelm the model and distract from the main task. Therefore, we have placed a strong emphasis on standardizing the input and output of this layer to facilitate the model's comprehension of the task objectives. Additionally, we strive to balance the simplicity of the input while ensuring comprehensive understanding.

Specifically, the instructions for each task are as follows:

| |
| --- |
| **INSTRUCTION:** Conceptual similarity judgment |
| **QUERY:** Among Tutu Chengcui, The Pierre, Waddesdon Manor, Astro Orbitor, Heian period, 2019 Canadian federal election, Paradiski, Tenughat Dam, Gros Michel banana, Reedy Glacier, Gangotri Glacier, Pinatubo, Interwar period, djon djon, Qiu Shiliang, Caciotta, Firth of Forth, 2011 Rugby World Cup, Cheng Yuanzhen, Pliocene, Sri Maha Bodhi, which one is the most conceptually similar with Botryosphaeria stevensii? Please answer the entity name only. |
| **ANSWER:** djon djon |

Table 8: The instruction and an example of Task 2-1 COPEN-CSJ, KU.

| |
| --- |
| **INSTRUCTION:** Conceptual property judgment |
| **QUERY:** Is the statement "Specieses have a cellulose wall and other polysaccharides." true or false? Please answer true or false. |
| **ANSWER:** False |

Table 9: The instruction and an example of Task 2-2 COPEN-CPJ, KU.

| |
| --- |
| **INSTRUCTION:** Conceptualization in contexts |
| **QUERY:** Given the context "The next year, he made his stock car racing debut in the American Speed Association, where he won a pole at Winchester Speedway and had four top-tens.", neglect your knowledge about Winchester Speedway and select the most contextually related concept for it from the concept set: Racecourse, Place, ArchitecturalStructure, Infrastructure, RaceTrack, Venue, Road, SportFacility, RouteOfTransportation, Building. Please answer the concept name only. |
| **ANSWER:** Racecourse |

Table 10: The instruction and an example of Task 2-3 COPEN-CiC, KU.

| |
|---|
| **INSTRUCTION:** Please recognize entities for the given text and classify them into a suitable type. The collection of types is as follows: <set of types> |
| **QUERY:** Agrippa succeeded in blocking the more manoeuvrable ships of Sextus and, after a long and bloody fight, to defeat his enemy. |
| **ANSWER:** Agrippa: person-politician; Sextus: person-politician; |

Table 11: The instruction and an example of Task 2-4 FewNERD, KU.

| |
|---|
| **INSTRUCTION:** Please follow the above demonstration, and extract relations from the [Question text]. Note the relation needs to be in the predefined set of relations. The output format required to is the same as the demonstration, format:(<entity_ID>, relation, <entity_ID>). The predefined set of relations: <set of types> |
| **QUERY:** <entity_0> Rickon Stark </entity_0> is a fictional character in the <entity_1> A Song of Ice and Fire </entity_1> series of fantasy novels by <entity_2> American </entity_2> author <entity_3> George R. R. Martin </entity_3>, and its television adaptation <entity_4> Game of Thrones </entity_4> . Introduced in <entity_5> 1996 </entity_5> 's <entity_6> A Game of Thrones </entity_6>, <entity_0> Rickon </entity_0> is the youngest child of <entity_7> Eddard Stark </entity_7>, the honorable lord of <entity_8> Winterfell </entity_8>, an ancient fortress in the <entity_9> North </entity_9> of the fictional kingdom of <entity_10> Westeros </entity_10>. He subsequently appeared in <entity_3> Martin </entity_3> 's <entity_11> A Clash of Kings </entity_11> (<entity_12> 1998 </entity_12>) . The <entity_13> Publishers Weekly </entity_13> review of <entity_6> A Game of Thrones </entity_6> noted, Ït is fascinating to watch <entity_3> Martin </entity_3> 's characters mature and grow, particularly <entity_14> Stark </entity_14> 's children, who stand at the center of the book. <̌entity_0> Rickon </entity_0> is played by <entity_15> Art Parkinson </entity_15> in the <entity_16> HBO </entity_16> television adaptation. |
| **ANSWER:** (<entity_0>, father, <entity_7>); (<entity_0>, present in work, <entity_4>); (<entity_0>, creator, <entity_3>); (<entity_7>, child, <entity_0>); (<entity_7>, present in work, <entity_6>); (<entity_7>, present in work, <entity_4>); (<entity_6>, publication date, <entity_5>); (<entity_6>, characters, <entity_7>); (<entity_6>, author, <entity_3>); (<entity_4>, publication date, <entity_5>); (<entity_4>, characters, <entity_7>); (<entity_4>, has part, <entity_6>); (<entity_4>, author, <entity_3>); (<entity_4>, screenwriter, <entity_3>); (<entity_4>, original network, <entity_16>); (<entity_3>, notable work, <entity_6>); (<entity_10>, present in work, <entity_4>); (<entity_11>, publication date, <entity_12>); (<entity_11>, series, <entity_6>); (<entity_11>, follows, <entity_6>); (<entity_11>, series, <entity_4>); (<entity_11>, author, <entity_3>); (<entity_14>, present in work, <entity_4>); (<entity_14>, creator, <entity_3>); |

Table 12: The instruction and an example of Task 2-5 DocRED, KU.

| |
|---|
| **INSTRUCTION:** Please identify the events in the text and classify them into appropriate categories; The collection of categories is <set of types> |
| **QUERY:** The ruling National Command of the Arab Socialist Ba'ath Party were removed from power by a union of the party's Military Committee and the Regional Command, under the leadership of Salah Jadid. |
| **ANSWER:** removed:Removing |

Table 13: The instruction and an example of Task 2-6 MAVEN, KU.

INSTRUCTION: Please classify the relation between two events/"Time" in a given document. There are 10 types of relations: ["before", "overlap", "contains", "simultaneous", "begins-on", "ends-on", "cause", "precondition", "subeven", and "coreference"]. In each document, 2 events/"Timex" are marked as "<Event> event name </Event>" or "<Timex> Timex name </Timex>". If there is a relation type or multiple relation types, the answer form is "Answer: [relation type 1, relation type 2, ...]".

QUERY: Document: The Central Park jogger case was a criminal case in the United States based on the assault and <Event> rape </Event> of Trisha Meili, a 28-year-old white woman who was jogging in the park, and attacks on eight other persons, in areas ranging from the North Woods of Manhattan's Central Park to the Reservoir, on the night of April 19, 1989. Three of the victims were black or Latino. Meili was so injured that she was in a coma for 12 days. "The New York Times" in 1990 described the attack on her as "one of the most widely publicized crimes of the 1980s". Attacks in Central Park that night were allegedly committed by a loose group of 30̌201332 teenagers, and police attempted to apprehend suspects after crimes began to be reported between 9 and 10 p.m. The brutally beaten Meili was not found until 1:30 a.m., after which the police hunt greatly intensified. They took into custody 14 or more other suspects over the next few days, and arrested a total of ten suspects who were ultimately tried for the attacks. Among them were four African American and two Hispanic American teenagers who were indicted on May 10 on charges of assault, robbery, riot, rape, sexual abuse, and attempted murder of Meili and an unrelated man, John Loughlin. The prosecutor planned to try the defendants in two groups, then scheduled the sixth defendant to be tried last. The latter pleaded guilty in January 1991 on lesser charges and received a reduced sentence. Prosecution of the five remaining defendants in the rape and assault case was based primarily on confessions which they had made after police interrogations. None had counsel during this questioning. Within weeks, they each withdrew these confessions, pleaded not guilty, and refused plea deals on the rape and assault charges. None of the suspects ' DNA matched the DNA collected from the crime scene: two semen samples that both belonged to one unidentified man. No substantive physical evidence connected any of the five teenagers to the rape scene, but each was convicted in 1990 of related assault and other charges. Subsequently known as the Central Park Five, they received stiff sentences ranging from 5 to 15 years. Four of the defendants appealed their convictions, but these were affirmed by appellate courts. The four juvenile defendants served 6̌20137 years each; the 16-year-old, tried and sentenced as an adult, served 13 years in adult prison. The five other defendants, <Event> indicted </Event> for assaults of other victims, pleaded guilty to reduced charges and received less severe sentences. In 2001, Matias Reyes, a convicted murderer and serial rapist serving life in prison, confessed to officials that he had raped the female jogger. His DNA matched that found at the scene, and he provided other confirmatory evidence. He said he committed the rape alone. Reyes could not be prosecuted for raping Meili, because the statute of limitations had passed. In 2002 Robert Morgenthau, District Attorney for New York County, had his office conduct an investigation and recommended to the state court that the convictions of the five men on all charges be vacated. The court vacated their convictions in 2002, and the state withdrew all charges against the men. In 2003, the five men sued the City of New York for malicious prosecution, racial discrimination, and emotional distress. The city refused to settle the suits for a decade, because its lawyers believed that the city could win a court case. After a change in administration, the city settled in 2014 with the five plaintiffs for $41 million. The five men also filed suit against the State of New York for additional damages; this case was settled in 2016 for a total of $3.9 million. The first event/"Timex": <Event> rape </Event>. The second event/"Timex": <Event> indicted </Event>.

ANSWER: before

Table 14: The instruction and an example of Task 2-7 MAVEN-ERE, KU.

INSTRUCTION: Please follow the above demonstration, extract relations from the [Question text]. Note the relation need to be in the predefined set of relations. The output format required to is the same as the demonstration, format:(<entity_ID>, relation, <entity_ID>). The predefined set of relations: <set of types>

QUERY: Text: Less than four months removed from the entity0 World Cup entity0's bright lights , the U.S. men's national soccer team visited an 8,000 - seat bayside stadium on a tiny entity1 Caribbean entity1 island Friday to face an opponent with players from regional leagues and some of entity2 England entity2's lowest divisions entity3. <text continued> Relations in the predefined set of relations in the above text:?

ANSWER: (entity4,part of,entity13); (entity6,part of,entity13); (entity13,has part,entity4); (entity15,participant of,entity0); (entity15,country of citizenship,entity9); (entity16,country of citizenship,entity9); (entity17,country of citizenship,entity9); (entity18,country of citizenship,entity9); (entity19,country of citizenship,entity9); (entity20,country of citizenship,entity9);

Table 15: The instruction and an example of Task 2-8 ETU, KU.

## D.4 KNOWLEDGE APPLYING TASKS

Knowledge Applying (KA) level naturally involves multi-hop reasoning in the form of question answering, which is suitable for large-scale models to perform inference. One major challenge is that the contexts required for these reasoning steps are not consistent. One particular aspect is the KoRC task and ETA task, which assume that the model has access to a corresponding knowledge base. The original dataset for KoRC confirms Wikidata as the knowledge base. However, in the Evolving data, many pieces of knowledge cannot be directly found. Therefore, we construct a virtual knowledge base using the annotated triples and generate questions based on this knowledge base.

Finally, we adopt the following task instructions:

| |
|---|
| **INSTRUCTION:** Please answer the following question. |
| **QUERY:** Jeremy Theobald and Christopher Nolan share what profession? |
| **ANSWER:** Jeremy Theobald is an actor and producer. Christopher Nolan is a director, producer, and screenwriter. Therefore, they both share the profession of being a producer. So the answer is: producer. |

Table 16: The instruction and an example of Task 3-1 HotpotQA, KA.

| |
|---|
| **INSTRUCTION:** Please answer the following question. |
| **QUERY:** Which film came out first, Blind Shaft or The Mask Of Fu Manchu? |
| **ANSWER:** Blind Shaft is a 2003 film. The Mask Of Fu Manchu is a 1932 film. So the answer is: The Mask Of Fu Manchu. |

Table 17: The instruction and an example of Task 3-2 2WikiMultihopQA, KA.

| |
|---|
| **INSTRUCTION:** Please answer the following question. |
| **QUERY:** When did the first large winter carnival take place in the city where CIMI_FM is licensed to broadcast? |
| **ANSWER:** CIMI_FM is licensed to broadcast in Quebec City. The first large winter carnival in Quebec City took place in 1894. So the answer is: 1894. |

Table 18: The instruction and an example of Task 3-3 MuSiQue, KA.

| |
|---|
| **INSTRUCTION:** Please answer the following question. |
| **QUERY:** When was Neville A. Stanton's employer founded? |
| **ANSWER:** The employer of Neville A. Stanton is University of Southampton. The University of Southampton was founded in 1862. So the answer is: 1862. |

Table 19: The instruction and an example of Task 3-4 KQA Pro, KA.

| |
|---|
| **INSTRUCTION:** You are given one document and one anonymized real-world entity with one or more mentions in the passage. Then we will ask your a question about this anonymized entity. The questions cannot be answered solely within the document or the background knowledge. Your task is to leverage world knowledge you have like Wikipedia or Wikidata as background knowledge combined with the given document to answer the question related to the anonymized entity. You must output all answers in the end. |
| **QUERY:** Allen is a county in the U.S. state of Ohio. As of the 2010 census, the population was 106,331. The county seat is Lima. The county was created in 1820 and organized in 1831. The county is named for Colonel [a human being], who was killed leading his men at the Battle of Frenchtown, during the War of 1812. It has also been claimed the county was named for Revolutionary War soldier Ethan Allen, but the weight of the evidence in favor of [the human being] led the General Assembly to declare in 1976 that the county was named for him. Allen comprises the Lima, OH Metropolitan Statistical Area, which is also part of the Lima - Van Wert - Wapakoneta, OH Combined Statistical Area. |
| **QUESTION::** Which place was this human being born? |
| **ANSWER:** Rockbridge County. |

Table 20: The instruction and an example of Task 3-5 KoRC, KA.

| |
|---|
| **INSTRUCTION:** You are given one document and one anonymized real-world entity with one or more mentions in the passage. Then we will ask your a question about this anonymized entity. The questions cannot be answered solely within the document or the background knowledge. Your task is to leverage world knowledge you have like Wikipedia or Wikidata as background knowledge combined with the given document to answer the question related to the anonymized entity. You must output all answers in the end. |
| **QUERY:** Six months after its New Shepard rocket suffered a failure during flight, Blue Origin said Friday its review of the incident pinpointed a problem with its engine nozzle and that it is expecting to return to flight "soon." In September, the rocket lifted off and flew for just over a minute before bright flames flashed from the booster and the capsule's emergency abort system kicked in, propelling it away from the rocket. The mission carried only science experiments; no one was on board, and no one was injured on the ground. In a statement Friday, Blue Origin, the space venture founded by Amazon executive chairman Jeff Bezos, said that it would refly the mission, again carrying scientific payloads. (Bezos owns [daily newspaper].) A flight with people could come later. The vehicle is designed to carry as many as six people to the edge of space and back on suborbital tourist trips that allow passengers to experience weightlessness and view the earth from above. In the statement, Blue Origin said its investigation, which was overseen by the Federal Aviation Administration and included members of the National Transportation Safety Board, concluded that the problem was caused by a failure of the engine nozzle, which experienced "temperatures that exceeded the expected and analyzed values of the nozzle material." Engineers are "implementing corrective actions, including design changes to the combustion chamber and operating parameters," the statement said. "Additional design changes to the nozzle have improved structural performance under thermal and dynamic loads." The FAA said in a statement that it is reviewing Blue Origin's mishap report but that the investigation remains open. "FAA approval is required to close the investigation and for the New Shepard system to return to flight." It was unclear how long that could take. While the booster was lost, the capsule and the 36 payloads it was carrying landed safely under parachutes and can fly again, Blue Origin said. The booster, which under normal circumstances falls back to Earth and touches down softly on a landing pad so that it can be reused, was a total loss. The company was able to recover all the debris from the rocket within the designated hazard area, it said. Bezos flew on the first flight with people in 2021. It had since flown five other missions with people on board, including one with Star Trek actor William Shatner and television commentator Michael Strahan. It has not flown since the September incident. |
| **QUESTION::** What is the home country of [daily newspaper]? |
| **ANSWER:** United States |

Table 21: The instruction and an example of Task 3-6 ETA, KA.

### D.5 KNOWLEDGE CREATING TASKS

The Knowledge Creation (KC) level is particularly unique as each task involves two generation processes. Here, we present the process that considers generating subsequent knowledge, which is the most informative. However, for direct generation, the instruction can be replaced with "Complete the following generate" without specifying the "TRIPLETS" item.

The example instructions of creating tasks are shown below:

---

**INSTRUCTION:** Complete the following texts and make sure to contain all the events provided.

**EVENTS:** ## Title: Death of Freddie Gray;### Known Events;#### Event Trigger: charges;##### Event Type: Judgment communication;##### Event Arguments; Agent: Marilyn Mosby; Patient: six police officers; Reason: the medical examiner's report ruled Gray's death a homicide;#### Event Trigger: stated;##### Event Type: Statement;##### Event Arguments; Speaker: The prosecutors; Message: they had probable cause to file criminal charges against the six police officers; Details: who were believed to be...

**GIVEN CONTEXT:** ### Text To Be Completed; On April 12, 2015, Freddie Carlos Gray, Jr., a 25-year-old black man, was arrested by the Baltimore Police Department for possessing what the police alleged was an illegal knife under Baltimore law.While being transported in a police van, Gray fell into a coma and was taken to a trauma center.Gray died on April 19, 2015 ; his death was ascribed to injuries to his spinal cord.On April 21, 2015, pending an investigation of the incident, six Baltimore police officers were suspended with pay...

**REFERENCE COMPLETION:** On May 1, 2015, the Baltimore City State's Attorney, Marilyn Mosby, announced her office had filed charges against six police officers after the medical examiner's report ruled Gray's death a homicide. The prosecutors stated that they had probable cause to file criminal charges against the six police officers who were believed to be involved in his death...

Table 22: The instruction and an example of Task 4-1 Encyclopedic, KC.

---

**INSTRUCTION:** Complete the following texts and make sure to contain all the events provided.

**EVENTS:** ## <Title if Contained>;### Known Events;#### Event Trigger: uncovered;##### Event Type: Reveal secret;##### Event Arguments; Speaker: Alston & Bird; Message: no facts to show that U.S. Soccer knew of the 1992 Incident when it hired Mr. Berhalter; Receiver: Gregg Berhalter;#### Event Trigger: harm;##### Event Type: Bodily harm;##### Event Arguments; Agent: Gregg Berhalter; Cause: others; Location: United States; ...

**GIVEN CONTEXT:** ### Text To Be Completed; Details of a sordid rift between two prominent U. S. soccer families — one that included allegations of domestic abuse against men' s national team coach Gregg Berhalter and parental complaints about Gio Reyna' s playing time at the 2022 World Cup — continued to spill out Monday when the findings of an independent investigation were released...

**REFERENCE COMPLETION:** Earnie Stewart left the job last month. Anthony Hudson, a World Cup assistant, is the interim coach. The next coach will begin preparing the U.S. team for the 2026 World Cup, which will take place in the United States, Mexico and Canada. Berhalter guided the United States for four years, leading a young squad to two regional championships and a place in the World Cup, where it finished second in group play and lost to the Netherlands in the round of 16...

Table 23: The instruction and an example of Task 4-2 ETC, KC.

# E   DETAILS OF RESULT INFERENCE

Given the instructions and test sets for each task, we evalute a total of 21 models in the first season of KoLA. Here, we present some of the deployment environments of the models that participated in our first season, as well as some specific solutions implemented during the evaluations.

## E.1   DEPLOYMENT ENVIRONMENT AND MODEL INFORMATION

The participating models in the evaluation include two types: closed-source models that return answers through API calls, and open-source models that are deployed directly for inference (with a temperature set to 0). Here, we primarily introduce the software and hardware environment used for deploying the models. We utilize the widely-used *PyTorch* and *transformers* library to load open-source models. The evaluation experiments are conducted on an Ubuntu 20.04.4 server equipped with 112 Intel Xeon(R) Platinum 8336C CPU cores, and graphic cards that contained 8 NVIDIA A100 SXM 80GB GPUs. Besides, The CUDA version is 11.4, the Python version is 3.10.0, the PyTorch version is 2.0.0 and the transformers version is 4.28.1.

Table 24 presents the features of the selected LLMs in the first and second season. For the open-source models, we deploy them using their official versions, with particular emphasis on the HuggingFace versions. As for the closed-source models, we utilize the various model APIs available as of May 15, 2023. We also conduct thorough checks and re-inferencing in case of any network-related errors.

Table 24: Selected LLMs. Instruct and |Context| correspond to whether the model is with instruction tuning and the input context's length limitation. * indicates the size has not been officially confirmed.

| Model | Size | Type | Instruct | |Context| | Website Url |
|---|---|---|---|---|---|
| *Season 1 Open-Source Models* | | | | | |
| GPT-J | 6B | Open | w/o | 2,048 | https://huggingface.co/EleutherAI/gpt-j-6b |
| GPT-JT | 6B | Open | w/ | 2,048 | https://www.eleuther.ai/artifacts/gpt-j |
| GPT-NeoX | 20B | Open | w/o | 2,048 | https://www.eleuther.ai/artifacts/gpt- |
| BLOOM | 7B | Open | w/o | 2,048 | https://bigscience.huggingface.co/blog/bloom |
| T0++ | 11B | Open | w/ | 512 | https://huggingface.co/bigscience/T0 |
| LLaMa | 65B | Open | w/o | 2,048 | https://github.com/facebookresearch/llama |
| Alpaca | 7B | Open | w/ | 2,048 | https://crfm.stanford.edu/2023/03/13/alpaca.html |
| UL2 | 20B | Open | w/o | 512 | https://huggingface.co/google/ul2 |
| FLAN-T5 | 11B | Open | w/ | 512 | https://github.com/google-research/FLAN |
| FLAN-UL2 | 20B | Open | w/ | 2,048 | https://www.yitay.net/blog/flan-ul2-20b |
| GLM | 130B | Open | w/o | 2,048 | https://github.com/THUDM/GLM-130B |
| ChatGLM | 6B | Open | w/ | 2,048 | https://github.com/THUDM/ChatGLM-6B |
| *Season 1 Closed-Source Models* | | | | | |
| ChatGLM | 130B | API | w/ | 2,048 | Not Publicly Availabile, Comming Soon |
| GPT-3 curie v1 | 6.7B | API | w/o | 2,048 | https://platform.openai.com/overview |
| GPT-3 davinci v1 | 175B | API | w/o | 2,048 | https://platform.openai.com/overview |
| InstructGPT curie v1 | 6.7B* | API | w/ | 2,048 | https://platform.openai.com/overview |
| InstructGPT davinci v2 | 175B* | API | w/ | 2,048 | https://platform.openai.com/overview |
| GPT3.5-turbo | * | API | w/ | 2,048 | https://platform.openai.com/overview |
| GPT-4 | * | API | w/ | 2,048 | https://platform.openai.com/overview |
| Cohere-command | 52.4B | API | w/ | 4,096 | https://docs.cohere.com/docs/the-command-model |
| J2-Jumbo-Instruct | 178B* | API | w/ | 8,192 | https://www.ai21.com/blog/introducing-j2 |
| *Season 2 New Models* | | | | | |
| LLaMa2-chat | 7B | Open | w/ | 3,500 | https://huggingface.co/meta-llama/Llama-2-7b-chat-hf |
| Dolly-v2 | 7B | Open | w/ | 2,048 | https://huggingface.co/databricks/dolly-v2-12b |
| Vicuna | 13B | Open | w/ | 2,048 | https://huggingface.co/lmsys/vicuna-13b-v1.5 |
| RedPajama-Instruct | 7B | Open | w/ | 1,024 | https://huggingface.co/togethercomputer/RedPajama-INCITE-7B-Instruct |
| Tulu | 7B | Open | w/ | 2,048 | https://huggingface.co/allenai/tulu-7b |
| Chatglm2-32k | 6B | Open | w/ | 31,500 | https://huggingface.co/THUDM/chatglm2-6b-32k |
| Internlm-chat-8k | 7B | Open | w/ | 7,500 | https://huggingface.co/internlm/internlm-chat-7b-8k |

## E.2   SOLUTION FOR RUN-TIME EXCEPTIONS

Apart from issues such as user permissions and network environment when invoking the model API, the main challenges we encountered during the model evaluation process were limited input length for some models and output inconsistencies with the required format. Therefore, we have devised the following strategies to handle these exceptional cases during evaluation:

**Over-length Issue**: Due to the length limitations of certain models, performing 5-shot zero-shot inference becomes challenging for tasks with lengthy instructions. Therefore, we have devised the following strategies to enable the models to produce desired outputs: a) Reduce the number of examples until the input-output length requirements are met; b) If reducing the number of examples to one still fails to meet the requirements of all cases, skip the non-compliant cases and treat them as 0; c) If a model skips a substantial number of examples (over 90%) on a particular task, consider it a failure on that task and record it as "–".

**Disregarding Instruction**: Another factor that significantly affects the model's performance is its potential inability to comprehend the task instructions, resulting in failure to produce outputs in the specified format. This poses challenges for evaluating the generation-based open-ended answers. Therefore, we attempt to extract key information from the answers using techniques such as regular expressions and perform fuzzy matching. Unfortunately, there are still many scenarios where certain models fail to generate correct and valid answers. For models that fail to provide reasonable answers on over 90% of the test cases for a particular task, we mark them as N/A.

**Sensitive Result**: The models may also trigger or bypass their safety mechanisms in certain tasks, such as refusing to provide answers (due to mistakenly perceiving the given information as containing unsafe content) or generating sensitive content. For cases where the model refuses to provide an answer, we conduct manual checks to ensure that the data in the test set does not contain explicit, violent, discriminatory, or other inappropriate content. In the secure test set, for situations where the model refuses to answer or exhibits similar behavior, we handle them in the same way as mentioned above. For models that cannot provide reasonable answers on over 90% of the test cases for a particular task, we mark them as N/A.

In general, for all tasks, if a model's performance is marked as "–" or "N/A" on a particular task, we do not include that score in the calculation of the standard score. However, when calculating the overall rankings of the models across different levels, we consider these cases as "missing" and assign them a score of 0. There is still room for improvement in this handling approach. Nonetheless, we firmly believe that the model's ability to handle input length, adhere to guidelines, and handle sensitive information is an important foundational skill for dealing with real-world knowledge problems.

Table 25: The Task Adaption Parameters in KoLA (Season 1st). "Max Train" and "Max Eval" correspond to the maximum number of examples and test cases. "Output" corresponds to the output format. We both include necessary and additional parameters required to fully specify evaluation (i.e. the number of evaluation instances and runs, which influence the statistical validity and reliability of the results), though they are not strictly part of the adaptation process, as HELM (Liang et al., 2022) presents.

| Level | ID | Dataset | Temperature | Max Tokens | Stop sequence(s) | Max Train | Max Eval | Output |
|-------|-----|---------|-------------|------------|------------------|-----------|----------|--------|
| KM | 1-1 | High-Freq. | 1 | 64 | None | 5 | 100 | List |
|    | 1-2 | Low-Freq. | 1 | 64 | None | 5 | 100 | |
|    | 1-3 | ETM | 1 | 64 | None | 5 | 100 | List |
| KU | 2-1 | COPEN-CSJ | 1 | 128 | None | 5 | 100 | |
|    | 2-2 | COPEN-CPJ | 1 | 128 | None | 5 | 100 | |
|    | 2-3 | COPEN-CiC | 1 | 128 | None | 5 | 100 | |
|    | 2-4 | FewNERD | 1 | 128 | None | 5 | 100 | String |
|    | 2-5 | DocRED | 1 | 256 | \n | 4 | 100 | |
|    | 2-6 | MAVEN | 0.2 | 200 | None | 5 | 100 | |
|    | 2-7 | MAVEN-ERE | 1 | 200 | None | 5 | 100 | |
|    | 2-8 | ETU | 1 | 256 | \n | 4 | 100 | String |
| KA | 3-1 | HotpotQA | 1 | 128 | \n | 5 | 100 | |
|    | 3-2 | 2WikiMulti. | 1 | 128 | \n | 5 | 100 | |
|    | 3-3 | MuSiQue | 1 | 128 | \n | 5 | 100 | String |
|    | 3-4 | KQA Pro | 1 | 128 | \n | 5 | 100 | |
|    | 3-5 | KoRC | 1 | 50 | <stop> | 5 | 100 | |
|    | 3-6 | ETA | 1 | 20 | <stop> | 5 | 100 | String |
| KC | 4-1 | Encyclopedic | 0 | 256 | None | 5 | 100 | String |
|    | 4-2 | ETC | 0 | 256 | None | 5 | 100 | String |

### E.3 ANSWER NORMALIZATION AND SCORING IMPLEMENTATION

Besides, for different tasks, we adjust some parameters used in the inference process based on the model's overall performance, as detailed in Table 25. Once we obtain the model inference results, we select various evaluation metrics and answer normalization methods according to the task's characteristics.

**Inference Implementation**. Following the approach of HELM (Liang et al., 2022), we first designed a unified dataset format template to store instructions, provided examples, test inputsoutputs, and decoding parameters. These parameters, akin to HELM, are adapted based on the specific task, as detailed in Table 25. After model inference, we document the following information: a) the text completed by model inference; b) the duration of the inference; c) the timestamp of the inference; d) each token inferred by the model, and f) the probability of each token after log_softmax, which is used for further analysis and score computation.

**Answer Normalization**. Since all tasks are set up as open-ended generative question-answering formats, to best reflect the model's true performance, we select different evaluation metrics tailored to the attributes of various task levels for post-processing the answers. Specifically, for the tasks that utilize F1 as the major evaluation metric, we employ a relaxed F1 (token match rate), i.e., after tokenizing the model's predicted results and the reference answers using the GPT2Tokenizer (Radford et al., 2019), we compute whether each token in the prediction is contained in the corresponding position of the gold standard. For the tasks are formed as classification tasks, we employ Accuracy of the final result as the major evaluation metric, while the generative tasks utilize the widely-accepted Rouge (Lin, 2004). We also make this portion of the code publicly available in our github repository, for introducing more detailed tricks to lift the reproducibility.

**Mechanism for Standardized Scores**. To make the model scores more comparable, we calculate the standardized scores by pairing each Evolving Task with its corresponding Known Task within the same level, such as 2-5 with 2-8, and 3-5 with 3-6. Notably, the three tasks at the first level are computed together. After determining the standardized scores for each level, we then compute the average standardized score for each level. The final overall score is derived from the average of these average standardized scores across levels. In this manner, we aim to mitigate potential biases caused by the varying number of tasks across different levels.

# F   MORE EVALUATION RESULT (SEASON 1)

Due to the length constraints of the paper, including the manual annotation process in the experimental section, a series of specific results are not fully presented. In this section, we first introduce the annotation process and results for the knowledge creation level. Then, we provide a detailed list of all absolute performance values for each model in the first season and discuss some notable findings.

## F.1   ANNOTATION OF CREATING TASKS

**Annotation Team for Evaluation**. We recruit members from the annotation team who are willing to participate in result annotation and further confirm their compensation and bonuses to protect their earnings and rights. Currently, 14 annotators participate in the annotation of knowledge creation results. The composition of these individuals closely resembles that of the overall annotation team, with all members having a bachelor's degree or higher. The annotation results undergo quality checks conducted by three Ph.D. students from KoLA teams, ensuring both efficiency and quality in the annotation process. We use Streamlit[18] to build an annotation platform where annotators can simultaneously view the text to be annotated, the context, and the historical annotations.

Table 26: Statistics of the annotation team for creating evaluation in KoLA.

| Gender | Rate |
|---|---|
| Female | 71.4% |
| Male | 28.6% |

| Education | Rate |
|---|---|
| Bachelor | 57.1% |
| Master | 42.9% |

**Annotation Process and Result**. For each model, we provide the following information: a) preceding context, b) continuation generated by the model, c) event knowledge content of the actual subsequent text, and d) actual subsequent text. It is important to note that in item b), the model generates the continuation without being informed of the knowledge in the subsequent text. Annotators are required to evaluate two aspects: i) the overall quality of the text, which includes assessing the novelty, coherence, and fluency of the model's creative content; and ii) the plausibility of the generated content in terms of knowledge hallucination, ensuring it does not conflict with real-world situations. The latter aspect is a subtask of the former. The annotation is conducted on a 5-point scale, where 1 represents the worst and 5 represents the best. We collect 4.2K scoring results and calculate the average score for each model. Table 27 presents the results and variances of the two ratings for each model. It is based on these scores and the model's rank that we calculate various correlation coefficients.

Table 27: The human evaluation results of Knowledge Creating (KC).

| Model | Overall | Knowledge | Model | Overall | Knowledge | Model | Overall | Knowledge |
|---|---|---|---|---|---|---|---|---|
| FLAN-T5 | 2.25 | 3.00 | LLaMa | 2.25 | 2.75 | InstructGPT curie v1 | 1.75 | 2.75 |
| UL2 | 2.00 | 2.25 | Alpaca | 1.25 | 2.00 | InstructGPT davinci v2 | 2.50 | 2.75 |
| FLAN-UL2 | 3.00 | 2.00 | J2-Jumbo-Instruct | 2.50 | 3.00 | GLM | 2.75 | 3.00 |
| GPT-J | 2.00 | 2.25 | Cohere-command | 2.75 | 3.00 | GPT-4 | 2.75 | 3.75 |
| GPT-NeoX | 3.25 | 3.25 | GPT-3.5-turbo | 2.50 | 3.00 | ChatGLM | 2.50 | 2.00 |
| BLOOM | 3.25 | 2.00 | GPT-3 curie v1 | 1.75 | 3.50 | ChatGLM (130B) | 2.50 | 3.00 |
| T0++ | 2.25 | 2.75 | GPT-3 davinci v1 | 2.50 | 2.50 | GPT-JT | 2.50 | 2.25 |

---

[18]https://streamlit.io/

Table 28: **Season 1**'s standardized performance of Knowledge Memorization and Understanding level.

| Model | Level 1: KM | | | | Level 2: KU | | | | | | | | |
|---|---|---|---|---|---|---|---|---|---|---|---|---|---|
| | 1-1 | 1-2 | 1-3 | Rank | 2-1 | 2-2 | 2-3 | 2-4 | 2-5 | 2-6 | 2-7 | 2-8 | Rank |
| GPT-4 | 50.1 | 54.0 | 53.1 | 1st | 63.3 | 42.5 | 45.2 | 60.7 | 100.0 | 70.9 | 72.7 | 59.4 | 1st |
| GPT-3.5-turbo | 41.2 | 46.6 | 41.4 | 4th | 38.0 | 43.2 | 44.1 | 47.8 | 47.0 | 44.0 | 50.4 | 25.3 | 2nd |
| InstructGPT davinci v2 (175B*) | 31.1 | 37.0 | 32.6 | 8th | 27.5 | 42.1 | 36.2 | 35.6 | 52.9 | 56.1 | 34.5 | 31.0 | 3rd |
| Cohere-command (52.4B) | 45.8 | 42.0 | 55.2 | 2nd | 33.8 | 40.9 | 40.2 | 20.5 | 33.3 | 14.6 | 40.7 | 18.4 | 4th |
| FLAN-UL2 (20B) | 40.8 | 32.2 | 51.6 | 5th | 52.8 | 40.9 | 46.9 | 9.9 | 18.4 | 14.6 | 16.1 | 18.4 | 6th |
| FLAN-T5 (11B) | 43.4 | 39.5 | 48.5 | 3rd | 57.0 | 41.7 | 43.0 | 12.6 | — | — | — | — | 5th |
| Tulu (7B) | 31.3 | 37.2 | 41.1 | 6th | 15.9 | 19.3 | 36.8 | 29.6 | 24.8 | 14.6 | 16.1 | 19.0 | 12th |
| J2-Jumbo-Instruct (178B*) | 23.9 | 24.8 | 18.9 | 12th | 21.2 | 17.4 | 25.1 | 31.0 | 26.1 | 26.5 | 45.1 | 21.8 | 7th |
| ChatGLM (130B) | 28.4 | 43.7 | 36.0 | 7th | 24.4 | 41.7 | 45.7 | 9.9 | 18.4 | 17.2 | 24.2 | 18.4 | 8th |
| InstructGPT curie v1 (6.7B*) | 20.2 | 33.2 | 33.2 | 9th | 23.3 | 35.0 | 35.7 | 16.2 | 18.9 | 15.9 | 19.7 | 18.4 | 9th |
| LLaMa (65B) | 17.0 | 18.1 | 11.8 | 14th | 15.9 | 12.3 | 12.2 | 49.4 | 25.4 | 24.0 | 19.4 | 18.4 | 11th |
| Llama2-chat (7B) | 14.8 | 13.2 | 13.9 | 19th | 19.1 | 12.3 | 18.4 | 31.2 | 26.6 | 14.6 | 21.1 | 18.9 | 14th |
| Chatglm2-32k (6B) | 17.9 | 15.8 | 10.6 | 16th | 15.9 | 34.6 | 13.9 | 12.9 | 20.0 | 14.6 | 16.7 | 19.0 | 17th |
| T0++ (11B) | 31.6 | 28.7 | 26.0 | 10th | 24.4 | 33.1 | 21.7 | 9.9 | — | — | — | — | 16th |
| Alpaca (7B) | 14.7 | 17.8 | 12.8 | 15th | 15.9 | 12.3 | 12.8 | 19.2 | 20.0 | 25.2 | 16.1 | 18.4 | 20th |
| GLM (130B) | 15.0 | 17.7 | 11.3 | 17th | 15.9 | 12.3 | 12.2 | 43.4 | 27.4 | 23.6 | 16.1 | 27.4 | 10th |
| Vicuna (13B) | 12.4 | 12.7 | 11.3 | 26th | 15.9 | 12.7 | 17.2 | 23.7 | 19.5 | 29.3 | 24.3 | 18.4 | 15th |
| UL2 (20B) | 18.9 | 20.2 | 12.7 | 13th | 15.9 | 12.3 | 12.2 | 11.9 | — | — | — | — | 28th |
| Dolly-v2 (12B) | 14.4 | 14.3 | 12.2 | 22th | 15.9 | 12.3 | 12.8 | 22.4 | 18.4 | 14.6 | 21.0 | 18.4 | 23th |
| ChatGLM (6B) | 23.3 | 24.2 | 20.7 | 11th | 17.0 | 39.3 | 26.7 | 9.9 | 18.4 | 15.9 | 16.7 | 18.4 | 13th |
| GPT-J (6B) | 14.1 | 12.4 | 10.9 | 24th | 15.9 | 12.3 | 12.2 | 16.1 | 18.4 | 24.9 | — | 18.4 | 25th |
| GPT-3 davinci v1 (175B) | 11.9 | 11.7 | 10.5 | 27th | 15.9 | 12.7 | 12.2 | 24.1 | 18.4 | 23.5 | 16.3 | 18.4 | 19th |
| Internlm-chat-8k (7B) | 16.3 | 13.8 | 11.6 | 20th | 15.9 | 12.3 | 12.2 | 13.0 | 20.8 | 14.6 | 20.0 | 18.5 | 27th |
| GPT-JT (6B) | 13.3 | 12.6 | 11.2 | 25th | 15.9 | 12.3 | 12.2 | 13.1 | 18.4 | 30.5 | — | 18.4 | 22th |
| GPT-NeoX (20B) | 13.4 | 14.0 | 11.0 | 23th | 15.9 | 12.3 | 12.2 | 19.6 | 18.4 | 25.9 | — | 18.4 | 21th |
| BLOOM (7B) | 14.3 | 15.1 | 13.0 | 18th | 15.9 | 12.3 | 12.2 | 24.0 | 21.7 | 23.1 | 16.1 | 18.4 | 18th |
| GPT-3 curie v1 (6.7B) | 11.1 | 11.6 | 10.5 | 28th | 15.9 | 12.3 | 12.2 | 15.4 | 18.4 | 19.6 | 17.9 | 18.4 | 26th |
| RedPajama-Instruct (7B) | 15.0 | 14.5 | 12.3 | 21th | 15.9 | 12.3 | 12.2 | 23.7 | 18.4 | 14.6 | 19.3 | 18.4 | 24th |

Table 29: **Season 1**'s standardized performance of Knowledge Applying, Creating level and all the 4 levels.

| Model | Level 3: KA | | | | | | | Level 4: KC | | | Overall (1,2,3,4) | |
|---|---|---|---|---|---|---|---|---|---|---|---|---|
| | 3-1 | 3-2 | 3-3 | 3-4 | 3-5 | 3-6 | Rank | 4-1 | 4-2 | Rank | Avg | Rank |
| GPT-4 | 53.6 | 54.5 | 70.6 | 26.7 | 54.1 | 53.3 | 1st | 44.8 | 50.0 | 3rd | 2.32 | 1st |
| GPT-3.5-turbo | 52.3 | 36.4 | 47.7 | 39.5 | 24.5 | 24.1 | 4th | 48.7 | 52.0 | 2nd | 1.51 | 2nd |
| InstructGPT davinci v2 (175B*) | 24.5 | 33.6 | 38.1 | 17.5 | 43.6 | 42.4 | 6th | 51.1 | 50.7 | 1st | 1.20 | 3rd |
| Cohere-command (52.4B) | 30.1 | 35.5 | 39.1 | 43.1 | 48.0 | 51.6 | 3rd | 10.9 | 33.7 | 12th | 0.90 | 4th |
| FLAN-UL2 (20B) | 43.4 | 41.4 | 33.3 | 45.0 | 37.0 | 49.4 | 2nd | 22.9 | 14.5 | 17th | 0.67 | 5th |
| FLAN-T5 (11B) | 38.8 | 42.8 | 26.8 | 45.0 | 33.2 | — | 7th | 14.2 | 16.2 | 22th | 0.49 | 6th |
| Tulu (7B) | 36.4 | 39.4 | 34.5 | 35.8 | 10.0 | | 5th | 27.8 | 25.4 | 11th | 0.48 | 7th |
| J2-Jumbo-Instruct (178B*) | 39.1 | 25.5 | 25.5 | 32.2 | 22.0 | 14.8 | 8th | 39.5 | 40.6 | 4th | 0.42 | 8th |
| ChatGLM (130B) | 30.2 | 27.9 | 21.9 | 30.3 | 30.2 | 10.0 | 10th | 18.8 | 16.9 | 18th | 0.19 | 9th |
| InstructGPT curie v1 (6.7B*) | 25.5 | 31.0 | 18.2 | 23.0 | 24.8 | 25.6 | 11th | 22.4 | 21.5 | 15th | 0.09 | 10th |
| LLaMa (65B) | 10.4 | 29.3 | 35.6 | 19.3 | 15.5 | 18.3 | 14th | 40.6 | 30.7 | 5th | 0.02 | 11th |
| Llama2-chat (7B) | 11.5 | 10.8 | 17.3 | 8.3 | 34.0 | 43.6 | 16th | 26.6 | 32.3 | 6th | -0.17 | 12th |
| Chatglm2-32k (6B) | 29.6 | 25.0 | 18.0 | 34.0 | 14.5 | 13.2 | 12th | 29.6 | 25.9 | 8th | -0.20 | 13th |
| T0++ (11B) | 16.4 | 16.9 | 17.7 | 8.3 | 33.2 | — | 19th | 12.0 | 23.5 | 19th | -0.20 | 14th |
| Alpaca (7B) | 9.0 | 13.2 | 14.8 | 12.0 | 38.9 | 25.3 | 17th | 31.0 | 22.6 | 10th | -0.29 | 15th |
| GLM (130B) | 17.4 | 6.9 | 12.3 | 15.7 | 38.4 | 38.4 | 13th | 24.0 | 15.0 | 16th | -0.30 | 16th |
| Vicuna (13B) | 19.3 | 3.9 | 18.6 | 12.0 | 14.8 | 11.4 | 23th | 30.1 | 28.7 | 7th | -0.36 | 17th |
| UL2 (20B) | 19.1 | 21.9 | 19.8 | 32.2 | 10.0 | — | 18th | 21.5 | 22.6 | 14th | -0.39 | 18th |
| Dolly-v2 (12B) | 8.1 | 14.4 | 12.3 | 12.0 | 20.1 | 18.2 | 22th | 24.2 | 30.0 | 9th | -0.41 | 19th |
| ChatGLM (6B) | 15.1 | 21.4 | 16.1 | 13.8 | 13.3 | 18.6 | 20th | 11.5 | 13.0 | 24th | -0.42 | 20th |
| GPT-J (6B) | 32.7 | 33.2 | 20.7 | 43.1 | 11.3 | 12.7 | 9th | 25.3 | 1.4 | 23th | -0.48 | 21th |
| GPT-3 davinci v1 (175B) | 12.3 | 7.5 | 16.7 | 12.0 | 15.3 | 11.5 | 25th | 27.3 | 16.8 | 13th | -0.57 | 22th |
| Internlm-chat-8k (7B) | 13.2 | 14.7 | 16.7 | 8.3 | 10.9 | 14.5 | 24th | 20.7 | 10.8 | 20th | -0.67 | 23th |
| GPT-JT (6B) | 25.3 | 32.0 | 16.9 | 26.7 | 12.3 | 13.0 | 15th | 15.4 | 0.0 | 26th | -0.67 | 24th |
| GPT-NeoX (20B) | 7.9 | 7.9 | 11.0 | 12.0 | 16.4 | 13.0 | 27th | 27.2 | 3.8 | 21th | -0.70 | 25th |
| BLOOM (7B) | 12.5 | 16.5 | 11.0 | 13.8 | 21.1 | 17.3 | 21th | 11.8 | 3.3 | 27th | -0.74 | 26th |
| GPT-3 curie v1 (6.7B) | 16.5 | 9.1 | 14.4 | 12.0 | 12.9 | 10.1 | 26th | 18.0 | 6.2 | 25th | -0.79 | 27th |
| RedPajama-Instruct (7B) | 6.6 | 3.9 | 11.0 | 8.3 | 19.8 | 17.5 | 28th | 7.3 | 0.2 | 28th | -0.91 | 28th |

## F.2 STANDARD RESULTS

Table 28 and Table 29 present the standard performance of models on the data of Season 1, which is the initial data annotated before June, 2023. The tested models exhibit a different rank with the ranking result in Season 2, which shows the changing feature of our evolving dataset.

## F.3 DETAILED RESULTS OF EACH TASK

**Knowledge Memorization (KM).** Table 30 present the absolute performance on the tasks of knowledge memorization. This level employs two main scoring methods, namely Exactly Match (EM) and Token-level F1. A key observation is that due to the limited control over generation by many models, the scores obtained using EM are often lower, resulting in numerous cases where a score cannot be assigned. During the analysis of the results, we observe that some models, even without access to external resources, can achieve good performance on evolving task. However, this often requires a substantial scale and instruction tuning. This may be attributed to the fact that certain new knowledge can be inferred from existing knowledge, which also relies on the model's memorization.

Table 30: Absolute Performance of all metrics on task (1-1), (1-2), and (1-3), KM.

| Model | 1-1 | | 1-2 | | 1-3 | |
|---|---|---|---|---|---|---|
| | EM | F1 | EM | F1 | EM | F1 |
| FLAN-T5 (11B) | 13.6 | 20.1 | 12.5 | 17.7 | 21.8 | 23.2 |
| UL2 (20B) | N/A | 5.1 | N/A | 5.9 | N/A | 1.4 |
| FLAN-UL2 (20B) | 14.0 | 18.5 | 9.5 | 13.2 | 23.7 | 25.1 |
| GPT-JT (6B) | N/A | 1.8 | N/A | 1.3 | N/A | 0.4 |
| GPT-J (6B) | N/A | 2.3 | N/A | 1.2 | N/A | 0.3 |
| GPT-NeoX (20B) | N/A | 1.8 | N/A | 2.2 | N/A | 0.3 |
| BLOOM (7B) | N/A | 2.3 | N/A | 2.8 | N/A | 1.5 |
| T0++ (11B) | 7.5 | 12.9 | 6.0 | 11.1 | 4.9 | 9.5 |
| LLaMa (65B) | 0.7 | 4.0 | N/A | 4.7 | N/A | 0.8 |
| Alpaca (7B) | N/A | 2.6 | 1.0 | 4.5 | N/A | 1.5 |
| J2-Jumbo-Instruct (178B*) | 4.6 | 8.2 | 5.0 | 8.8 | 1.0 | 5.2 |
| Cohere-command (52.4B) | 17.0 | 21.5 | 12.5 | 19.3 | 24.7 | 27.3 |
| GPT-3.5-turbo | 9.8 | 18.7 | 18.0 | 22.1 | 13.8 | 18.9 |
| GPT-3 curie v1 (6.7B) | N/A | 0.4 | N/A | 0.7 | N/A | N/A |
| GPT-3 davinci v1 (175B) | N/A | 0.9 | N/A | 0.8 | N/A | N/A |
| InstructGPT curie v1 (6.7B*) | 1.3 | 5.9 | 10.5 | 13.9 | 7.9 | 13.9 |
| InstructGPT davinci v2 (175B*) | 5.6 | 12.6 | 13.0 | 16.2 | 9.6 | 13.5 |
| GLM (130B) | N/A | 2.8 | N/A | 4.4 | N/A | 0.5 |
| GPT-4 | 17.1 | 24.2 | 20.8 | 26.5 | 21.0 | 26.0 |
| ChatGLM (130B) | 7.4 | 10.9 | 16.5 | 20.3 | 13.8 | 15.6 |

**Knowledge Understanding (KU).** The results in this level are unexpected, as a significant amount of knowledge understanding relies on longer texts or generating highly structured content. Therefore, in complex tasks such as document-level relation extraction and event relation extraction, the performance of many models is not satisfactory. This aspect deserves further exploration.

Table 31: Absolute Performance of accuracy on COPEN (2-1), (2-2), (2-3), KU.

| Model | 2-1 | 2-2 | 2-3 |
|---|---|---|---|
| FLAN-T5 (11B) | 39.0 | 75.0 | 55.0 |
| UL2 (20B) | N/A | N/A | N/A |
| FLAN-UL2 (20B) | 35.0 | 73.0 | 62.0 |
| GPT-JT (6B) | N/A | N/A | N/A |
| GPT-J (6B) | N/A | N/A | N/A |
| GPT-NeoX (20B) | N/A | N/A | N/A |
| BLOOM (7B) | N/A | N/A | N/A |
| T0++ (11B) | 8.0 | 53.0 | 17.0 |
| LLaMa (65B) | N/A | N/A | N/A |
| Alpaca (7B) | N/A | N/A | 1.0 |
| J2-Jumbo-Instruct (178B*) | 5.0 | 13.0 | 23.0 |
| Cohere-command (52.4B) | 17.0 | 73.0 | 50.0 |
| GPT-3.5-turbo | 21.0 | 79.0 | 57.0 |
| GPT-3 curie v1 (6.7B) | N/A | N/A | N/A |
| GPT-3 davinci v1 (175B) | N/A | 1.0 | N/A |
| InstructGPT curie v1 (6.7B*) | 7.0 | 58.0 | 42.0 |
| InstructGPT davinci v2 (175B*) | 11.0 | 76.0 | 43.0 |
| GLM (130B) | N/A | N/A | N/A |
| GPT-4 | 45.0 | 77.0 | 59.0 |
| ChatGLM (130B) | 8.0 | 75.0 | 60.0 |

Table 32: Absolute Performance on RE tasks, i.e., FewNERD (2-4), DocRED (2-5), ETU (2-8), KU.

| Model | 2-4 | | | 2-5 | | | 2-8 | | |
|---|---|---|---|---|---|---|---|---|---|
| | P | R | F1 | P | R | F1 | P | R | F1 |
| FLAN-T5 (11B) | 7.8 | 0.4 | 0.7 | — | — | — | — | — | — |
| UL2 (20B) | 16.7 | 0.3 | 0.5 | — | — | — | — | — | — |
| FLAN-UL2 (20B) | N/A | N/A | N/A | N/A | N/A | N/A | N/A | N/A | N/A |
| GPT-JT (6B) | 20.0 | 0.4 | 0.8 | N/A | N/A | N/A | N/A | N/A | N/A |
| GPT-J (6B) | 1.9 | 1.7 | 1.6 | N/A | N/A | N/A | N/A | N/A | N/A |
| GPT-NeoX (20B) | 2.6 | 2.5 | 2.5 | N/A | N/A | N/A | N/A | N/A | N/A |
| BLOOM (7B) | 3.3 | 4.6 | 3.7 | 1.0 | 1.8 | 1.3 | N/A | N/A | N/A |
| T0++ (11B) | N/A | N/A | N/A | — | — | — | — | — | — |
| LLaMa (65B) | 10.0 | 11.1 | 10.4 | 2.2 | 3.8 | 2.8 | N/A | N/A | N/A |
| Alpaca (7B) | 2.1 | 3.2 | 2.4 | 0.6 | 0.7 | 0.6 | N/A | N/A | N/A |
| J2-Jumbo-Instruct (178B*) | 5.7 | 5.5 | 5.5 | 3.0 | 3.0 | 3.0 | 1.5 | 1.2 | 1.3 |
| Cohere-command (52.4B) | 5.2 | 1.9 | 2.8 | 5.2 | 6.6 | 5.8 | N/A | N/A | N/A |
| GPT-3.5-turbo | 10.4 | 9.5 | 10.0 | 11.9 | 10.6 | 11.2 | 4.1 | 2.0 | 2.7 |
| GPT-3 curie v1 (6.7B) | 1.3 | 1.7 | 1.4 | N/A | N/A | N/A | N/A | N/A | N/A |
| GPT-3 davinci v1 (175B) | 3.6 | 3.9 | 3.7 | N/A | N/A | N/A | N/A | N/A | N/A |
| InstructGPT curie v1 (6.7B*) | 2.4 | 1.3 | 1.6 | 0.3 | 0.2 | 0.2 | N/A | N/A | N/A |
| InstructGPT davinci v2 (175B*) | 6.0 | 7.9 | 6.8 | 13.2 | 13.9 | 13.6 | 5.3 | 4.6 | 5.0 |
| GLM (130B) | 7.9 | 10.5 | 8.8 | 2.8 | 4.6 | 3.5 | 2.8 | 4.6 | 3.5 |
| GPT-4 | 12.4 | 14.5 | 13.4 | 35.5 | 29.2 | 32.0 | 19.8 | 13.6 | 16.1 |
| ChatGLM (130B) | N/A | N/A | N/A | N/A | N/A | N/A | N/A | N/A | N/A |

Table 33: Absolute Performance of all metrics on two sub-tasks of MAVEN (2-6), KU.

| Model | 2-6 | | | | | |
| --- | --- | --- | --- | --- | --- | --- |
| | Identification | | | Classification | | |
| | P | R | F1 | P | R | F1 |
| FLAN-T5 (11B) | — | — | — | — | — | — |
| UL2 (20B) | — | — | — | — | — | — |
| FLAN-UL2 (20B) | N/A | N/A | N/A | N/A | N/A | N/A |
| GPT-JT (6B) | 21.8 | 12.1 | 15.6 | 15.1 | 8.4 | 10.8 |
| GPT-J (6B) | 25.0 | 8.4 | 12.5 | 13.9 | 4.7 | 7.0 |
| GPT-NeoX (20B) | 11.8 | 12.6 | 12.2 | 7.5 | 7.9 | 7.7 |
| BLOOM (7B) | 30.6 | 8.8 | 13.7 | 12.9 | 3.7 | 5.8 |
| T0++ (11B) | — | — | — | — | — | — |
| LLaMa (65B) | 52.9 | 16.7 | 25.4 | 13.2 | 4.2 | 6.4 |
| Alpaca (7B) | 48.6 | 7.9 | 13.6 | 25.7 | 4.2 | 7.2 |
| J2-Jumbo-Instruct (178B*) | 28.0 | 10.7 | 15.5 | 14.6 | 5.6 | 8.1 |
| Cohere-command (52.4B) | N/A | N/A | N/A | N/A | N/A | N/A |
| GPT-3.5-turbo | 54.7 | 34.9 | 42.6 | 25.5 | 16.3 | 19.9 |
| GPT-3 curie v1 (6.7B) | 52.2 | 5.6 | 10.1 | 17.4 | 1.9 | 3.4 |
| GPT-3 davinci v1 (175B) | 44.2 | 10.7 | 17.2 | 15.4 | 3.7 | 6.0 |
| InstructGPT curie v1 (6.7B*) | 100.0 | 0.5 | 0.9 | 100.0 | 0.5 | 0.9 |
| InstructGPT davinci v2 (175B*) | 52.4 | 45.1 | 48.5 | 30.3 | 26.0 | 28.0 |
| GLM (130B) | 52.5 | 19.5 | 28.5 | 11.2 | 4.2 | 6.1 |
| GPT-4 | 66.3 | 58.6 | 62.2 | 40.5 | 35.8 | 38.0 |
| ChatGLM (130B) | 50.0 | 2.3 | 4.4 | 20.0 | 0.9 | 1.8 |

Table 34: Absolute Performance of all metrics on four sub-tasks of MAVEN-ERE (2-7), KU.

| Model | 2-7 | | | | | | | | | | | |
| --- | --- | --- | --- | --- | --- | --- | --- | --- | --- | --- | --- | --- |
| | Temporal | | | Causal | | | Subevent | | | Coreference | | |
| | P | R | F1 | P | R | F1 | P | R | F1 | P | R | F1 |
| FLAN-T5 (11B) | — | — | — | — | — | — | — | — | — | — | — | — |
| UL2 (20B) | — | — | — | — | — | — | — | — | — | — | — | — |
| FLAN-UL2 (20B) | N/A | N/A | N/A | N/A | N/A | N/A | N/A | N/A | N/A | N/A | N/A | N/A |
| GPT-JT (6B) | — | — | — | — | — | — | — | — | — | — | — | — |
| GPT-J (6B) | — | — | — | — | — | — | — | — | — | — | — | — |
| GPT-NeoX (20B) | — | — | — | — | — | — | — | — | — | — | — | — |
| BLOOM (7B) | N/A | N/A | N/A | N/A | N/A | N/A | N/A | N/A | N/A | N/A | N/A | N/A |
| T0++ (11B) | — | — | — | — | — | — | — | — | — | — | — | — |
| LLaMa (65B) | N/A | N/A | N/A | 33.3 | 5.0 | 8.7 | N/A | N/A | N/A | N/A | N/A | N/A |
| Alpaca (7B) | N/A | N/A | N/A | N/A | N/A | N/A | N/A | N/A | N/A | N/A | N/A | N/A |
| J2-Jumbo-Instruct (178B*) | 10.7 | 3.4 | 5.2 | 25.0 | 30.0 | 27.3 | 22.2 | 20.0 | 21.1 | 25.0 | 20.0 | 22.2 |
| Cohere-command (52.4B) | 7.5 | 6.9 | 7.2 | 15.0 | 15.0 | 15.0 | N/A | N/A | N/A | 44.4 | 40.0 | 42.1 |
| GPT-3.5-turbo | 18.6 | 14.9 | 16.6 | 15.7 | 55.0 | 24.4 | 4.0 | 10.0 | 5.7 | 33.3 | 60.0 | 42.9 |
| GPT-3 curie v1 (6.7B) | 4.7 | 4.6 | 4.7 | N/A | N/A | N/A | N/A | N/A | N/A | N/A | N/A | N/A |
| GPT-3 davinci v1 (175B) | 0.3 | 1.1 | 0.5 | N/A | N/A | N/A | N/A | N/A | N/A | N/A | N/A | N/A |
| InstructGPT curie v1 (6.7B*) | 6.9 | 14.9 | 9.4 | N/A | N/A | N/A | N/A | N/A | N/A | N/A | N/A | N/A |
| InstructGPT davinci v2 (175B*) | 22.0 | 14.9 | 17.8 | 11.8 | 30.0 | 16.9 | 20.0 | 10.0 | 13.3 | N/A | N/A | N/A |
| GLM (130B) | N/A | N/A | N/A | N/A | N/A | N/A | N/A | N/A | N/A | N/A | N/A | N/A |
| GPT-4 | 19.0 | 26.4 | 22.1 | 23.8 | 25.0 | 24.4 | 21.4 | 30.0 | 25.0 | 72.7 | 80.0 | 76.2 |
| ChatGLM (130B) | 18.7 | 3.4 | 5.8 | N/A | N/A | N/A | N/A | N/A | N/A | 33.3 | 10.0 | 15.4 |

**Knowledge Applying (KA).** In the evaluation of the KA level, a notable phenomenon is that knowledge graph (KG)-based reasoning question answering tasks are almost impossible to complete without the use of KG. This phenomenon is evident in the three tasks (3-4)-(3-6). Furthermore, due to the clear quality progression exhibited by multiple tasks in this layer, the performance of models generally follows a decreasing trend.

Table 35: Absolute Performance of F1-score on task (3-1), (3-2) and (3-3), KA.

| Model | 3-1 | 3-2 | 3-3 |
|---|---|---|---|
| FLAN-T5 (11B) | 23.7 | 35.3 | 7.5 |
| UL2 (20B) | 9.2 | 16.3 | 4.2 |
| FLAN-UL2 (20B) | 27.1 | 34.0 | 10.6 |
| GPT-JT (6B) | 13.8 | 25.5 | 2.8 |
| GPT-J (6B) | 19.2 | 26.6 | 4.6 |
| GPT-NeoX (20B) | 1.0 | 3.6 | N/A |
| BLOOM (7B) | 4.4 | 11.4 | N/A |
| T0++ (11B) | 7.2 | 11.8 | 3.2 |
| LLaMa (65B) | 2.8 | 23.0 | 11.7 |
| Alpaca (7B) | 1.8 | 8.4 | 1.8 |
| J2-Jumbo-Instruct (178B*) | 23.9 | 19.6 | 6.9 |
| Cohere-command (52.4B) | 17.3 | 28.7 | 13.4 |
| GPT-3.5-turbo | 33.6 | 29.5 | 17.5 |
| GPT-3 curie v1 (6.7B) | 7.3 | 4.7 | 1.6 |
| GPT-3 davinci v1 (175B) | 4.2 | 3.2 | 2.7 |
| InstructGPT curie v1 (6.7B*) | 13.9 | 24.6 | 3.4 |
| InstructGPT davinci v2 (175B*) | 13.2 | 26.9 | 12.9 |
| GLM (130B) | 8.0 | 2.7 | 0.6 |
| GPT-4 | 34.6 | 45.9 | 28.4 |
| ChatGLM (130B) | 17.4 | 21.8 | 5.2 |

Table 36: Absolute Performance of accuracy of different types of questions on KQA Pro (3-4), KA.

| Model | 3-4 | | | | | | |
|---|---|---|---|---|---|---|---|
| | All | Multi. | Quali. | Comp. | Logi. | Count. | Veri. |
| FLAN-T5 (11B) | 20.0 | 21.3 | 19.4 | 29.4 | 15.4 | N/A | 64.3 |
| UL2 (20B) | 13.0 | 13.3 | 22.6 | 17.6 | 15.4 | N/A | 14.3 |
| FLAN-UL2 (20B) | 20.0 | 21.3 | 19.4 | 41.2 | 19.2 | N/A | 42.9 |
| GPT-JT (6B) | 10.0 | 12.0 | 19.4 | N/A | 15.4 | N/A | 28.6 |
| GPT-J (6B) | 19.0 | 17.3 | 22.6 | 35.3 | 19.2 | N/A | 42.9 |
| GPT-NeoX (20B) | 2.0 | 2.7 | N/A | N/A | N/A | N/A | 14.3 |
| BLOOM (7B) | 3.0 | 4.0 | 6.5 | N/A | 3.8 | N/A | 7.1 |
| T0++ (11B) | N/A | N/A | N/A | N/A | N/A | N/A | N/A |
| LLaMa (65B) | 6.0 | 8.0 | 12.9 | 5.9 | 11.5 | N/A | 7.1 |
| Alpaca (7B) | 2.0 | 2.7 | N/A | 5.9 | N/A | N/A | N/A |
| J2-Jumbo-Instruct (178B*) | 13.0 | 12.0 | 12.9 | 17.6 | 3.8 | 11.1 | 28.6 |
| Cohere-command (52.4B) | 19.0 | 21.3 | 22.6 | 23.5 | 23.1 | 11.1 | 50.0 |
| GPT-3.5-turbo | 17.0 | 16.0 | 19.4 | 41.2 | 19.2 | N/A | 21.4 |
| GPT-3 curie v1 (6.7B) | 2.0 | 1.3 | 3.2 | 5.9 | 3.8 | N/A | N/A |
| GPT-3 davinci v1 (175B) | 3.0 | 2.7 | 3.2 | N/A | N/A | N/A | 14.3 |
| InstructGPT curie v1 (6.7B*) | 8.0 | 8.0 | 3.2 | 29.4 | 7.7 | N/A | 14.3 |
| InstructGPT davinci v2 (175B*) | 5.0 | 4.0 | 6.5 | 11.8 | 7.7 | N/A | N/A |
| GLM (130B) | 4.0 | 5.3 | 6.5 | N/A | 7.7 | N/A | 7.1 |
| GPT-4 | 10.0 | 12.0 | 16.1 | 5.9 | 7.7 | N/A | 42.9 |
| ChatGLM (6B) | 3.0 | 2.7 | 6.5 | 5.9 | 3.8 | N/A | N/A |
| ChatGLM (130B) | 12.0 | 10.7 | 9.7 | 35.3 | 7.7 | N/A | 21.4 |

Table 37: Absolute Performance of all metrics on KoRC (3-5) and ETA (3-6), KA.

| Model | 3-5 | | 3-6 | |
|---|---|---|---|---|
| | EM | F1 | EM | F1 |
| FLAN-T5 (11B) | 20.0 | 23.3 | — | — |
| UL2 (20B) | N/A | N/A | — | — |
| FLAN-UL2 (20B) | 21.0 | 27.1 | 32.6 | 39.6 |
| GPT-JT (6B) | N/A | 2.3 | N/A | 3.0 |
| GPT-J (6B) | N/A | 1.4 | N/A | 2.7 |
| GPT-NeoX (20B) | 2.0 | 6.5 | N/A | 3.1 |
| BLOOM (7B) | 7.0 | 11.2 | N/A | 7.3 |
| T0++ (11B) | 18.0 | 23.4 | — | — |
| LLaMa (65B) | 2.0 | 5.6 | N/A | 8.4 |
| Alpaca (7B) | 23.0 | 29.1 | 4.1 | 15.4 |
| J2-Jumbo-Instruct (178B*) | 6.0 | 12.1 | 2.0 | 4.9 |
| Cohere-command (52.4B) | 28.0 | 38.1 | 36.7 | 41.8 |
| GPT-3.5-turbo | 10.0 | 14.6 | 10.2 | 14.2 |
| GPT-3 curie v1 (6.7B) | 2.0 | 3.0 | N/A | 0.2 |
| GPT-3 davinci v1 (175B) | 3.0 | 5.3 | N/A | 1.5 |
| InstructGPT curie v1 (6.7B*) | 9.0 | 14.9 | 8.2 | 15.7 |
| InstructGPT davinci v2 (175B*) | 25.0 | 33.8 | 22.4 | 32.6 |
| GLM (130B) | 22.0 | 28.5 | 22.0 | 28.5 |
| GPT-4 | 33.0 | 44.3 | 36.7 | 43.5 |
| ChatGLM (6B) | 2.0 | 3.3 | N/A | 8.7 |
| ChatGLM (130B) | 17.0 | 20.3 | N/A | N/A |

**Knowledge Creating (KC).** Here, we present the scores of three sub-criteria used to calculate the overall score for each model. If only these scores are considered, it is observed that some well-regarded models such as GPT4 and GPT-3.5-turbo do not necessarily demonstrate superiority.

Table 38: Absolute Performance of the key metrics on Encyclopedia Creating task (4-1), KC.

| Model | $\partial(T_k, R)$ | $\partial(T, R)$ | $\partial(T, T_k)$ |
|---|---|---|---|
| FLAN-T5 (11B) | 15.6 | 9.7 | 22.5 |
| UL2 (20B) | 23.3 | 18.3 | 31.6 |
| FLAN-UL2 (20B) | 17.3 | 8.7 | 14.5 |
| GPT-JT (6B) | 22.1 | 17.8 | 36.0 |
| GPT-J (6B) | 27.5 | 19.4 | 32.9 |
| GPT-NeoX (20B) | 28.2 | 19.2 | 31.6 |
| BLOOM (7B) | 25.4 | 19.5 | 44.6 |
| T0++ (11B) | 16.3 | 10.0 | 25.7 |
| LLaMa (65B) | 30.9 | 24.0 | 25.5 |
| Alpaca (7B) | 26.5 | 20.0 | 26.9 |
| J2-Jumbo-Instruct (178B*) | 27.5 | 17.8 | 17.0 |
| Cohere-command (52.4B) | 27.6 | 19.3 | 47.5 |
| GPT-3.5-turbo | 45.6 | 21.3 | 29.2 |
| GPT-3 curie v1 (6.7B) | 26.5 | 22.1 | 42.0 |
| GPT-3 davinci v1 (175B) | 28.0 | 21.7 | 33.7 |
| InstructGPT curie v1 (6.7B*) | 20.2 | 18.8 | 28.0 |
| InstructGPT davinci v2 (175B*) | 43.7 | 22.4 | 26.0 |
| GLM (130B) | 30.1 | 20.3 | 37.7 |
| GPT-4 | 37.9 | 14.8 | 19.0 |
| ChatGLM (6B) | 20.0 | 15.4 | 35.3 |
| ChatGLM (130B) | 17.5 | 16.0 | 26.1 |

Table 39: Absolute Performance of the key metrics on ETC (4-2), KC.

| Model | $\partial(T_k, R)$ | $\partial(T, R)$ | $\partial(T, T_k)$ |
|---|---|---|---|
| FLAN-T5 (11B) | 1.0 | 7.3 | 3.6 |
| UL2 (20B) | 1.2 | 15.6 | 5.6 |
| FLAN-UL2 (20B) | 9.1 | 6.1 | 12.1 |
| GPT-JT (6B) | 17.4 | 14.9 | 43.9 |
| GPT-J (6B) | 17.7 | 16.0 | 44.1 |
| GPT-NeoX (20B) | 18.0 | 16.5 | 42.3 |
| BLOOM (7B) | 19.5 | 16.4 | 44.3 |
| T0++ (11B) | 0.9 | 15.8 | 4.5 |
| LLaMa (65B) | 26.0 | 20.0 | 26.6 |
| Alpaca (7B) | 14.9 | 20.7 | 24.5 |
| J2-Jumbo-Instruct (178B*) | 27.0 | 17.8 | 15.4 |
| Cohere-command (52.4B) | 26.1 | 15.0 | 18.6 |
| GPT-3.5-turbo | 45.7 | 19.1 | 23.9 |
| GPT-3 curie v1 (6.7B) | 18.9 | 17.4 | 41.7 |
| GPT-3 davinci v1 (175B) | 24.0 | 20.3 | 39.0 |
| InstructGPT curie v1 (6.7B*) | 20.8 | 21.1 | 31.8 |
| InstructGPT davinci v2 (175B*) | 43.1 | 21.3 | 24.7 |
| GLM (130B) | 25.8 | 17.1 | 39.4 |
| GPT-4 | 43.8 | 17.5 | 22.3 |
| ChatGLM (6B) | 17.7 | 14.9 | 31.1 |
| ChatGLM (130B) | 10.0 | 8.5 | 13.0 |

# G MORE EVALUATION RESULT (SEASON 2)

Due to the length constraints of the paper, the results of season 2 are not fully presented. In this section, we provide a detailed list of all absolute performance values for each model in the second season and discuss some notable findings.

## G.1 DETAILED RESULTS OF ROLLING TASKS FOR ALL MODELS

Table 40: Absolute performance of all metrics on rolling tasks: ETM (2-8), ETU (2-8), ETA (3-6) and ETC (4-2), **Season 2**.

| Model | 1-3 | | 2-8 | | | 3-6 | | 4-2 | | |
|---|---|---|---|---|---|---|---|---|---|---|
| | EM | F1 | P | R | F1 | EM | F1 | $\partial\left(T_k, R\right)$ | $\partial\left(T, R\right)$ | $\partial\left(T, T_k\right)$ |
| FLAN-T5 (11B) | 6.9 | 8.4 | N/A | N/A | N/A | N/A | N/A | 0.8 | 5.1 | 25.1 |
| UL2 (20B) | N/A | 1.3 | N/A | N/A | N/A | N/A | N/A | 0.0 | 13.7 | 23.2 |
| FLAN-UL2 (20B) | 5.7 | 7.2 | N/A | N/A | N/A | 35.0 | 36.8 | 1.6 | 4.7 | 4.8 |
| GPT-JT (6B) | N/A | 1.3 | N/A | N/A | N/A | N/A | 1.4 | 9.4 | 15.6 | 23.0 |
| GPT-J (6B) | N/A | 0.8 | N/A | N/A | N/A | N/A | 0.4 | 10.6 | 17.3 | 20.9 |
| GPT-NeoX (20B) | N/A | 0.9 | N/A | N/A | N/A | N/A | 0.6 | 17.1 | 17.8 | 33.3 |
| BLOOM (7B) | N/A | 1.0 | N/A | N/A | N/A | N/A | 7.1 | 24.6 | 26.3 | 46.9 |
| T0++ (11B) | 2.3 | 3.8 | N/A | N/A | N/A | N/A | N/A | 0.6 | 10.6 | 26.4 |
| LLaMa (65B) | N/A | 1.6 | N/A | N/A | N/A | N/A | 1.5 | 27.2 | 30.0 | 36.0 |
| Alpaca (7B) | N/A | 0.9 | N/A | N/A | N/A | 30.0 | 34.2 | 3.4 | 26.9 | 4.9 |
| J2-Jumbo-Instruct (178B*) | N/A | 1.4 | N/A | N/A | N/A | 5.0 | 9.3 | 30.8 | 29.0 | 21.0 |
| Cohere-command (52.4B) | 8.0 | 8.8 | N/A | N/A | N/A | 11.7 | 27.6 | 17.7 | 6.6 | 27.1 |
| GPT-3.5-turbo | 10.1 | 11.1 | 0.2 | 0.3 | 0.2 | N/A | 3.4 | 35.9 | 25.4 | 30.0 |
| GPT-3 curie v1 (6.7B) | N/A | 0.2 | N/A | N/A | N/A | N/A | 0.3 | 19.8 | 19.4 | 47.3 |
| GPT-3 davinci v1 (175B) | N/A | 0.2 | N/A | N/A | N/A | N/A | 0.9 | 22.7 | 22.0 | 39.4 |
| InstructGPT curie v1 (6.7B*) | 4.6 | 6.3 | N/A | N/A | N/A | 3.3 | 10.9 | 18.5 | 21.4 | 26.9 |
| InstructGPT davinci v2 (175B*) | 8.6 | 10.0 | 2.4 | 3.1 | 2.7 | N/A | 7.2 | 45.9 | 31.3 | 34.6 |
| GLM (130B) | N/A | 3.3 | — | — | — | 18.3 | 18.8 | 21.1 | 16.6 | 36.1 |
| GPT-4 | 10.5 | 12.7 | 22.7 | 25.5 | 24.1 | 40.0 | 41.5 | 47.7 | 23.1 | 25.9 |
| ChatGLM (130B) | 8.0 | 10.0 | N/A | N/A | N/A | N/A | 5.0 | 28.3 | 15.3 | 31.9 |
| ChatGLM (6B) | 5.7 | 7.1 | N/A | N/A | N/A | 5.0 | 12.2 | 21.3 | 16.9 | 23.8 |
| Dolly-v2 (12B) | N/A | 1.3 | N/A | N/A | N/A | N/A | 9.0 | 21.1 | 20.7 | 26.4 |
| RedPajama-Instruct (7B) | N/A | N/A | N/A | N/A | N/A | N/A | 7.0 | 15.7 | 15.3 | 36.7 |
| Tulu (7B) | 3.4 | 6.1 | 0.4 | 0.3 | 0.4 | 36.6 | 38.3 | 2.6 | 28.2 | 3.7 |
| Vicuna (13B) | N/A | 0.5 | N/A | N/A | N/A | N/A | 0.3 | 3.2 | 31.2 | 3.7 |
| Llama2-chat (7B) | N/A | 0.8 | 0.3 | 0.3 | 0.3 | 28.3 | 35.2 | 32.9 | 26.5 | 37.1 |
| Chatglm2-32k (6B) | 1.1 | 1.9 | 0.1 | 0.3 | 0.2 | N/A | 1.1 | 32.8 | 30.3 | 37.2 |
| Internlm-chat-8k (7B) | N/A | 0.4 | N/A | N/A | N/A | N/A | 6.1 | 24.0 | 21.3 | 34.3 |

Table 40 presents the absolute performance for all models on the rolling tasks with season 2's data. We find that most models still fail to get right results on 2-8 as the task require the model's ability on understanding the long context and complex structures.

## G.2 DETAILED RESULTS OF ALL TASKS FOR NEW MODELS

**Knowledge Memorization (KM).** Table 41 presents the absolute performance on the tasks of knowledge memorization with data from season 1 to test the 7 new models in season 2.

**Knowledge Understanding (KU).** The results in this level are similar with results in season 1, where most new models in season 2 are not able to get right results due to the long context and highly structured content.

**Knowledge Applying (KA).** Season 2's new models don't perform well on KQA Pro(3-4) tasks as shown in Table 47.

Table 41: Absolute Performance of all metrics for new models of Season 2 on task (1-1), (1-2), and (1-3), KM.

| Model | 1-1 | | 1-2 | | 1-3 | |
|---|---|---|---|---|---|---|
| | EM | F1 | EM | F1 | EM | F1 |
| Dolly-v2 (12B) | 0.5 | 2.4 | N/A | 2.4 | N/A | 1.1 |
| RedPajama-Instruct (7B) | N/A | 2.8 | N/A | 2.4 | N/A | 1.1 |
| Tulu (7B) | 9.3 | 12.7 | 9.5 | 16.3 | 17.9 | 18.7 |
| Vicuna (13B) | N/A | 1.2 | N/A | 1.4 | N/A | 0.5 |
| Llama2-chat (7B) | 1.0 | 2.7 | N/A | 1.7 | N/A | 2.1 |
| Chatglm2-32k (6B) | 2.1 | 4.5 | 1.0 | 3.2 | N/A | 0.1 |
| Internlm-chat-8k (7B) | 0.1 | 3.6 | N/A | 2.0 | N/A | 0.7 |

Table 42: Absolute Performance of accuracy for new models of Season 2 on COPEN (2-1), (2-2), (2-3), KU.

| Model | 2-1 | 2-2 | 2-3 |
|---|---|---|---|
| Dolly-v2 (12B) | N/A | N/A | 1.0 |
| RedPajama-Instruct (7B) | N/A | N/A | N/A |
| Tulu (7B) | N/A | 18.0 | 44.0 |
| Vicuna (13B) | N/A | 1.0 | 9.0 |
| Llama2-chat (7B) | 3.0 | N/A | 11.0 |
| Chatglm2-32k (6B) | N/A | 57.0 | 3.0 |
| Internlm-chat-8k (7B) | N/A | N/A | N/A |

Table 43: Absolute Performance for new models of Season 2 on RE tasks, i.e., FewNERD (2-4), DocRED (2-5), ETU (2-8), KU.

| Model | 2-4 | | | 2-5 | | | 2-8 | | |
|---|---|---|---|---|---|---|---|---|---|
| | P | R | F1 | P | R | F1 | P | R | F1 |
| Dolly-v2 (12B) | 3.1 | 3.6 | 3.3 | N/A | N/A | N/A | N/A | N/A | N/A |
| RedPajama-Instruct (7B) | 3.3 | 4.0 | 3.6 | N/A | N/A | N/A | N/A | N/A | N/A |
| Tulu (7B) | 7.4 | 4.0 | 5.2 | 2.2 | 2.9 | 2.5 | 3.3 | 0.1 | 0.2 |
| Vicuna (13B) | 3.2 | 4.3 | 3.6 | 0.5 | 0.4 | 0.4 | N/A | N/A | N/A |
| Llama2-chat (7B) | 4.4 | 7.8 | 5.6 | 3.5 | 3.0 | 3.2 | 0.2 | 0.2 | 0.2 |
| Chatglm2-32k (6B) | 0.7 | 0.9 | 0.8 | 0.6 | 0.7 | 0.6 | 0.2 | 0.3 | 0.2 |
| Internlm-chat-8k (7B) | 0.6 | 1.2 | 0.8 | 1.5 | 0.7 | 0.9 | N/A | 0.1 | N/A |

Table 44: Absolute Performance of all metrics for new models of Season 2 on two sub-tasks of MAVEN (2-6), KU.

| Model | 2-6 | | | | | |
|---|---|---|---|---|---|---|
| | Identification | | | Classification | | |
| | P | R | F1 | P | R | F1 |
| Dolly-v2 (12B) | N/A | N/A | N/A | N/A | N/A | N/A |
| RedPajama-Instruct (7B) | N/A | N/A | N/A | N/A | N/A | N/A |
| Tulu (7B) | N/A | N/A | N/A | N/A | N/A | N/A |
| Vicuna (13B) | 40.2 | 20.0 | 26.7 | 15.0 | 7.4 | 9.9 |
| Llama2-chat (7B) | N/A | N/A | N/A | N/A | N/A | N/A |
| Chatglm2-32k (6B) | N/A | N/A | N/A | N/A | N/A | N/A |
| Internlm-chat-8k (7B) | N/A | N/A | N/A | N/A | N/A | N/A |

Table 45: Absolute Performance of all metrics for new models of Season 2 on four sub-tasks of MAVEN-ERE (2-7), KU.

| Model | 2-7 | | | | | | | | | | | |
|---|---|---|---|---|---|---|---|---|---|---|---|---|
| | Temporal | | | Causal | | | Subevent | | | Coreference | | |
| | P | R | F1 | P | R | F1 | P | R | F1 | P | R | F1 |
| Dolly-v2 (12B) | 5.4 | 14.9 | 8.0 | 1.3 | 5.0 | 2.0 | 1.7 | 10.0 | 2.9 | N/A | N/A | N/A |
| RedPajama-Instruct (7B) | 3.2 | 6.9 | 4.4 | N/A | N/A | N/A | 1.1 | 10.0 | 1.9 | 1.2 | 10.0 | 2.2 |
| Tulu (7B) | N/A | N/A | N/A | N/A | N/A | N/A | N/A | N/A | N/A | N/A | N/A | N/A |
| Vicuna (13B) | 21.7 | 5.7 | 9.1 | 16.7 | 10.0 | 12.5 | N/A | N/A | N/A | N/A | N/A | N/A |
| Llama2-chat (7B) | 10.7 | 17.2 | 13.2 | N/A | N/A | N/A | N/A | N/A | N/A | N/A | N/A | N/A |
| Chatglm2-32k (6B) | 2.4 | 1.1 | 1.6 | N/A | N/A | N/A | N/A | N/A | N/A | N/A | N/A | N/A |
| Internlm-chat-8k (7B) | 3.7 | 4.6 | 4.1 | 3.4 | 25.0 | 6.1 | N/A | N/A | N/A | N/A | N/A | N/A |

Table 46: Absolute Performance of F1-score for new models of Season 2 on task (3-1), (3-2) and (3-3), KA.

| Model | 3-1 | 3-2 | 3-3 |
|---|---|---|---|
| Dolly-v2 (12B) | 1.1 | 9.5 | 0.6 |
| RedPajama-Instruct (7B) | N/A | N/A | N/A |
| Tulu (7B) | 21.9 | 32.2 | 11.2 |
| Vicuna (13B) | 9.4 | N/A | 3.6 |
| Llama2-chat (7B) | 3.6 | 6.2 | 3.0 |
| Chatglm2-32k (6B) | 16.9 | 19.1 | 3.3 |
| Internlm-chat-8k (7B) | 4.9 | 9.8 | 2.7 |

Table 47: Absolute Performance of accuracy of different types of questions for new models of Season 2 on KQA Pro (3-4), KA.

| Model | 3-4 | | | | | | |
|---|---|---|---|---|---|---|---|
| | All | Multi. | Quali. | Comp. | Logi. | Count. | Veri. |
| Dolly-v2 (12B) | 2.0 | 2.7 | 3.2 | N/A | N/A | N/A | 7.1 |
| RedPajama-Instruct (7B) | N/A | N/A | N/A | N/A | N/A | N/A | N/A |
| Tulu (7B) | 22.0 | 22.7 | 25.8 | 52.9 | 11.5 | 11.1 | 35.7 |
| Vicuna (13B) | 2.0 | 2.7 | N/A | 5.9 | N/A | N/A | N/A |
| Llama2-chat (7B) | N/A | N/A | N/A | N/A | N/A | N/A | N/A |
| Chatglm2-32k (6B) | 14.0 | 14.7 | 12.9 | 35.3 | 7.7 | N/A | 28.6 |
| Internlm-chat-8k (7B) | N/A | N/A | N/A | N/A | N/A | N/A | N/A |

Table 48: Absolute Performance of all metrics for new models of Season 2 on KoRC (3-5) and ETA (3-6), KA.

| Model | 3-5 | | 3-6 | |
|---|---|---|---|---|
| | EM | F1 | EM | F1 |
| Dolly-v2 (12B) | 5.0 | 10.2 | N/A | 8.2 |
| RedPajama-Instruct (7B) | 7.0 | 9.9 | N/A | 7.6 |
| Tulu (7B) | 19.0 | 25.9 | N/A | N/A |
| Vicuna (13B) | 2.0 | 4.9 | N/A | 1.4 |
| Llama2-chat (7B) | 18.0 | 24.1 | 26.5 | 33.8 |
| Chatglm2-32k (6B) | 3.0 | 4.6 | N/A | 3.3 |
| Internlm-chat-8k (7B) | N/A | 0.9 | N/A | 4.6 |

**Knowledge Creating (KC).** Here, we present the scores of three sub-criteria used to calculate the overall score for each new model in season 2.

Table 49: Absolute Performance of the key metrics for new models of Season 2 on Encyclopedia Creating task (4-1), KC.

| Model | $\partial(T_k, R)$ | $\partial(T, R)$ | $\partial(T, T_k)$ |
|---|---|---|---|
| Dolly-v2 (12B) | 26.8 | 19.8 | 33.8 |
| RedPajama-Instruct (7B) | 20.5 | 18.7 | 43.5 |
| Tulu (7B) | 3.7 | 16.8 | 4.1 |
| Vicuna (13B) | 2.9 | 18.7 | 2.8 |
| Llama2-chat (7B) | 20.2 | 21.4 | 26.3 |
| Chatglm2-32k (6B) | 26.2 | 18.7 | 26.6 |
| Internlm-chat-8k (7B) | 11.4 | 15.4 | 17.6 |

Table 50: Absolute Performance of the key metrics on ETC (4-2), KC.

| Model | $\partial(T_k, R)$ | $\partial(T, R)$ | $\partial(T, T_k)$ |
|---|---|---|---|
| Dolly-v2 (12B) | 24.9 | 21.7 | 28.0 |
| RedPajama-Instruct (7B) | 16.2 | 17.3 | 45.0 |
| Tulu (7B) | 0.1 | 14.1 | 0.1 |
| Vicuna (13B) | 24.9 | 20.5 | 28.0 |
| Llama2-chat (7B) | 0.7 | 21.4 | 1.0 |
| Chatglm2-32k (6B) | 23.6 | 17.8 | 26.9 |
| Internlm-chat-8k (7B) | 6.3 | 7.8 | 14.9 |

