# OpenReview forum: "KoLA: Carefully Benchmarking World Knowledge of Large Language Models"
_ICLR.cc/2024/Conference — ICLR 2024 poster_

### Official Review · Reviewer_vPns · 2023-10-28

**Soundness:** 2 fair
**Presentation:** 2 fair
**Contribution:** 2 fair
**Rating:** 5
**Confidence:** 2

**Summary:**

The paper studies how to thoroughly evaluating LLMs on its knowledge capability.  Inspired by cognitive science, the authors establish an extensive benchmark that focuses on memorization, understanding, applying and creating respectively. For each capability, the authors have chosen/created new tasks for that capability and evaluate a significant amount of LLMs to draw insights conclusions on those experiments.

The author also introduces a new metric for knowledge creating, which in the experiments shows a notable correlation to faithfulness.

**Strengths:**

The paper tackles a timely and important issues which evaluates the LLM capability instead of just evaluating on some tasks. To answer this question seems still hard and the paper has selected various datasets and made sensible grouping to evaluate the four aspect that it mentions.

The paper has run the experiments for several seasons for now and has shown some interesting trends that correlate with model size. The paper also proposes a novel metric for knowledge creation that seems interesting and notably correlated with faithfulness.

**Weaknesses:**

There are several questions that I think arise after reading the paper, I would consider these just missing some clarity:
The paper said that "Comparing the second-season results with the first season, the rankings of most open-source models have declined" but Table 2 and Table 3 seem to show results of four levels of the same season.
Why COPEN and 2wikiMultiQA are also considered as exclusive?

One of the potential strength of the paper is to analyse the results on the fresh data of each season that the paper claims; however, we don't find such results in the current version. Meanwhile, the paper draws some conclusions (the ones related to knowledge) that don't seem to be part of the contributions of this particular paper.

**Questions:**

The paper said that "Comparing the second-season results with the first season, the rankings of most open-source models have declined" but Table 2 and Table 3 seem to show results of four levels of the same season.

Why COPEN and 2wikiMultiQA are also considered as exclusive?

How the knowledge is designed in Figure 2 to apply self contrast metric please?

---

> ### Author Response · Authors · 2023-11-16
>
> Thank you for investing your time and expertise in reviewing our work. We are grateful for your recognition of our efforts in the design of framework and the evolving tests, and we are delighted to clarify the concerns and answer the questions you raised.
>
> - *The paper said that "Comparing the second-season results with the first season, the rankings of most open-source models have declined" but Table 2 and Table 3 seem to show results of four levels of the same season.*
>
> Due to space constraints, we placed the complete results of the first season in Appendix E.2. In the content, we primarily present the results of the second season (the latest season at the time of submission) and its comparison to the first season, which is reflected in the Rank column for each level.
>
> - *Why COPEN and 2wikiMultiQA are also considered as exclusive?*
>
> To ensure the reliability of our evaluation, we reached out to the authors of several works, including COPEN and 2WikiMutiQA, at the very beginning of our project. We obtained their **unpublished held-out test sets** for evaluation in KoLA. As explained in the caption of Table 1, “Exclusive tasks mean their test sets are newly developed or sponsored by the original authors and were not publicly disclosed.” Based on this criterion, we labeled these tasks as “Exclusive.”
>
> - *How the knowledge is designed in Figure 2 to apply self contrast metric please?*
>
> To more clearly demonstrate the evaluation process of the knowledge creation task, here we use the example in Figure 2 for a detailed explanation. The process can be divided into three main steps:
>
> First, the model to be evaluated generates a completion $T$ using only the context $C$ (The Battle of Evesham marked the defeat of Simon de Montfort...), allowing it to freely create content. Second, the model uses both the context $C$ and event knowledge $K$ (i.e., the structured content in the middle part of the figure) as inputs, to generate the other completion $T_{k}$. Third, by contrasting the model's generated results $T$ and $T_{k}$ under these two conditions, we can calculate the rationality of the model's event knowledge creation, $\partial \left( T, T_{k} \right)$ (its ability to freely create subsequent event knowledge, e.g., *Prince Edward’s Escape*). Together with the other two contrast metrics, this comparison ultimately leads to the scoring of the knowledge creation level. We hope this explanation can further aid your understanding of the paper.
>
> - *One of the potential strength of the paper is to analyse the results on the fresh data of each season that the paper claims; however, we don't find such results in the current version.*
>
> We analyzed the results on evolving tasks compared with known tasks in Section 3 (the second point of “Design Analysis”), which shows that the known-evolving performance gap is closer at the higher levels, indicating that the high-level abilities are more generalizable. We totally agree that a thorough analysis on evolving data is essential. We plan to execute a retrospective diachronic analysis after completing the first phase (six seasons as planned) and hope to have intriguing findings.
>
> - *Meanwhile, the paper draws some conclusions (the ones related to knowledge) that don't seem to be part of the contributions of this particular paper.*
>
> We believe that the experimental conclusions of this paper can provide contributions in two key aspects. First, the analysis upon a Bloom’s cognitive taxonomy, a rather unique aspect of this paper, offers a new perspective in analyzing the knowledge capabilities of LLMs. Second, although some speculations about model performance, such as the alignment tax, have been widely discussed, our results provide new experimental support for these speculations. As other reviewers have mentioned, although some conclusions might not be novel, the evaluation results provided by this framework are valuable. Additionally, KoLA, as a long-term maintained benchmark, can be utilized to observe whether these issues show improvement in the future.
>
> **[Final Remark]** Thank you again for reviewing our paper and for the pleased comments. We hope that our response and clarification have addressed your questions and concerns. We sincerely invite you to engage with us if you have more questions.

---

> > ### Comment · Reviewer_vPns · 2023-11-21
> > **Thank you for the detailed explanations**
> >
> > I appreciate the authors' detailed feedback. The main contribution of this work is the extensive benchmark with different levels (knowledge, reasoning, etc.) of data of different nature (evolving or not evolving, etc.) with inspiration drawn from cognitive science. However, although I agree with the novelty of this idea and the significant efforts authors have shown in the project as presented in the paper, I am not convinced after reading the paper that the hierarchy with the chosen benchmarks bring any benefits for 1) better understanding LLM capacity 2) the practical development or usage for LLMs. Thus I am not particularly excited of this work but I certainly have no strong arguments to reject this paper either.

---

> > > ### Author Response · Authors · 2023-11-22
> > >
> > > Thank you for taking the time to read our response and acknowledging our novelty and efforts. Considering that we focus on knowledge-related abilities, we understand that researchers from different backgrounds may have different judgments on the significance of this paper. Regardless, we believe that your valuable comments will be beneficial for our future work.

---

### Official Review · Reviewer_ryRT · 2023-11-01

**Soundness:** 3 good
**Presentation:** 3 good
**Contribution:** 2 fair
**Rating:** 6
**Confidence:** 4

**Summary:**

The paper presents a world knowledge benchmark for LLM, focusing on three aspects: (1) ability modeling; (2) evolving benchmark built upon emerging corpora and (3) evaluation criteria. The author also presented metrics of SOTA LLM's performance on the benchmark and provided insights observed from the evaluation results.

**Strengths:**

S1. The paper presents a new LLM benchmark with some innovations, including constructing benchmark with emerging corpora, evaluating model's knowledge creation capability.
S2. The paper evaluated major SOTA LLMs, providing good comparison in model's capability from different perspective.
S3. The paper reads well, easy to follow.

**Weaknesses:**

W1. Benchmark on emerging corpora is a great idea and it is quite encouraging to see the authors promised to refresh the benchmark regularly. However, it is not clear how to maintain such benchmark in the long term.
W2. It is not clear why we need another new LLM benchmark. Given all different benchmarks available publicly, I am not convinced KoLA is a must-have addition.
W3. It is not clear why the standardized overall scoring can give better idea than simple ranking.

**Questions:**

Q1. The paper identified four knowledge capabilities. It is clear on knowledge memorization and knowledge creation. However, it is vague to distinguish knowledge understanding and knowledge application. Take knowledge understanding as an example, would reading comprehension a task of knowledge understanding or knowledge attention? Why knowledge understanding only has extraction tasks?
Q2. Benchmark on emerging corpora is a great idea and it is quite encouraging to see the authors promised to refresh the benchmark regularly. However, it is not clear how to maintain such benchmark in the long term. As the time goes by, some evolving benchmark is no longer most up-to-date, how to handle the benchmark data? How would this evolving benchmark interact with the known dataset?
Q3. How much does the new KoLA benchmark differ from existing LLM benchmark? There are so many available benchmark published, each focusing on one/multiple capabilities of the model. Why do we need KoLA? What if we combined existing ones?
Q4. It is not clear the contribution/motivation with standardized overall scoring. As the score would depend on the evaluated models, it will change a lot as more evaluated models would be added in. Also why this is better than simple ranking?

---

> ### Author Response · Authors · 2023-11-16
>
> Thank you for dedicating your time and effort to review our work, and for acknowledging our overall design, experimental analysis, and writing. We are glad to answer the questions you've raised.
>
> - Q1: *The paper identified four knowledge capabilities. It is clear on knowledge memorization and knowledge creation. However, it is vague to distinguish knowledge understanding and knowledge application. Take knowledge understanding as an example, would reading comprehension a task of knowledge understanding or knowledge attention? Why knowledge understanding only has extraction tasks?*
>
> This is a valuable question, and we are pleased to discuss it with you.
>
> In Bloom's Cognitive Taxonomy [1], *Understanding* can be described as ”Determining the meaning of instructional messages“, typically involving actions like “Explain, Summarize.” *Applying* is described as “Carrying out or using a procedure in a given situation”, generally including actions such as 'Implement, Execute’. Indeed, since the revision of Bloom's theory [2], there has been considerable debate about the demarcation between these two cognitive levels.
>
> Despite ongoing discussions, one consensus principle to differentiate them is whether it involves information and abilities beyond the given content [3]. Consequently, we categorize tasks involving information extraction (which ensures that the output information is faithfully extracted from the input text) as Understanding. It is worth noting that the Knowledge Understanding (KU) level is not limited to information extraction; it also includes the ability of conceptual abstraction, as seen in our investigated task COPEN. Under this criterion, if a reading comprehension question requires only knowledge from the text, we would classify it as a KU task. Once it involves more complex background knowledge and reasoning, it might lean more towards a Knowledge Application (KA) task.
>
> Ref:
>
> [1] Krathwohl D R. A revision of Bloom's taxonomy: An overview[J]. Theory into practice, 2002, 41(4): 212-218.
>
> [2] Forehand M. Bloom's taxonomy: Original and revised[J]. Emerging perspectives on learning, teaching, and technology, 2005, 8: 41-44.
>
> [3] Wilson L O. Anderson and Krathwohl Bloom’s taxonomy revised understanding the new version of Bloom’s taxonomy[J]. The Second Principle, 2016, 1(1): 1-8.
>
> - Q2: *Benchmark on emerging corpora is a great idea and it is quite encouraging to see the authors promised to refresh the benchmark regularly. However, it is not clear how to maintain such benchmark in the long term. As the time goes by, some evolving benchmark is no longer most up-to-date, how to handle the benchmark data? How would this evolving benchmark interact with the known dataset?*
>
> As stated in Section 2 of our paper, we are committed to updating our evaluation for at least six seasons. After this, we will further extend tasks to introduce new versions of KoLA. The data for every new season will be up-to-date to ensure the test data is not leaked, maintaining the relative fairness of the evaluation. For the ended seasons, we do not plan to add them to the known set immediately. Instead, they will be publicly released after the completion of six seasons, and a retrospective analysis of all past seasons will be conducted. The primary workload for running a new season includes collecting and annotating new data, constructing a new list of models, and publishing evaluation results. To host these tasks, we have established a maintenance team and carefully controlled costs. This includes adding a large amount of automated pre-annotation (as described in Appendix B) and controlling the size of the evaluation set. As mentioned in our discussion with reviewer 1m8m, we expect the cost of every season will be under USD 2,000. Therefore, we are confident to fulfill the committed maintenance and we also welcome contributions from the community.

---

> > ### Author Response · Authors · 2023-11-16
> > **Official Comment by Authors (Part II)**
> >
> > - Q3: *How much does the new KoLA benchmark differ from existing LLM benchmark? There are so many available benchmark published, each focusing on one/multiple capabilities of the model. Why do we need KoLA? What if we combined existing ones?*
> >
> > We believe KoLA is unique in two key aspects. (1) As described in Section 1, rather than assessing the breadth of model capabilities like most existing benchmarks (such as MMLU), KoLA focuses more on analyzing the deep interconnections of model knowledge capabilities through carefully-designed multi-level testing. (2) Besides combining some existing tasks, KoLA maintains a unique evolving mechanism, aimed at minimizing data leakage. Recent studies [4,5,6] have widely revealed a serious issue of 'test set contamination' in the evaluation of many LLMs with existing benchmarks, where the results may not genuinely reflect the actual capabilities of the models. Therefore, we believe KoLA holds a distinctive value in the current landscape.
> >
> > Ref:
> >
> > [4] Wei T, Zhao L, Zhang L, et al. Skywork: A More Open Bilingual Foundation Model[J]. arXiv preprint arXiv:2310.19341, 2023.
> >
> > [5] Yang S, Chiang W L, Zheng L, et al. Rethinking Benchmark and Contamination for Language Models with Rephrased Samples[J]. arXiv preprint arXiv:2311.04850, 2023.
> >
> > [6] Zhou K, Zhu Y, Chen Z, et al. Don't Make Your LLM an Evaluation Benchmark Cheater[J]. arXiv preprint arXiv:2311.01964, 2023.
> >
> > - Q4: *It is not clear the contribution/motivation with standardized overall scoring. As the score would depend on the evaluated models, it will change a lot as more evaluated models would be added in. Also why this is better than simple ranking?*
> >
> > Our considerations about employing standardized scoring are as follows: Standardized scores have become the mainstream choice in fields like educational assessment and intelligence testing [7], with systems like the SAT, GRE, and Wechsler Intelligence Scales (IQ Test) adopting them. Compared to absolute values, using standard scores in the multi-task evaluation of LLMs offers the following benefits:
> >
> > 1. Standardized scores enable fair comparisons across different seasons and datasets. The varying difficulty of tasks and the different sensitivities of metrics mean that the absolute numerical performance of models is incomparable. For example, if there is a dataset where models typically perform at 80, and another where the usual performance is 40, directly comparing a model's performance scores on these two datasets cannot reflect whether the model is better at one task over the other. However, comparing standardized scores derived from relative performance can yield conclusive insights.
> > 2. For overall rankings, standardized scores can negate the bias caused by specific tasks. For instance, without standardized scoring, ChatGLM (6B) would rank much higher in overall rankings solely due to its high performance in the 2-2 task, surpassing its improved version ChatGLM2-32k.
> >
> > We acknowledge that the inclusion of new models indeed results in variations in the overall standardized scores. We believe that considering the benefits mentioned above, such variations are acceptable. One of the objectives of KoLA is to provide a reference for users in selecting models. We trust that if new models demonstrate superior capabilities, the relative downscaling of other models' evaluations is beneficial for users.
> >
> > Ref:
> >
> > [7] Haladyna T M, Nolen S B, Haas N S. Raising standardized achievement test scores and the origins of test score pollution[J]. Educational Researcher, 1991, 20(5): 2-7.
> >
> > **[Final Remark]** Thank you again for reviewing our paper and for the insightful discussions. We hope that our response and clarification have addressed your questions and concerns. Please feel invited to engage with us if you have more questions.

---

### Official Review · Reviewer_VBLJ · 2023-11-03

**Soundness:** 3 good
**Presentation:** 3 good
**Contribution:** 4 excellent
**Rating:** 8
**Confidence:** 2

**Summary:**

The authors proposes a world knowledge assessment benchmark KoLA that consists of factual knowledge from Wikidata to evaluate the world knowledge of large language models (LLMs). KoLA consists of a four-level taxonomy: knowledge memorization/knowledge understanding/knowledge applying/knowledge creating. The first two tasks focus on directly extracting/generating the information associated with the corresponding world knowledge, and the last two focus on the application of knowledge in reasoning and text generation.

The data comes from both the data already exposed to LLMs (known data) and the data appeared afterwards (evolving data). Results show that pre-alignment/instruction-tuning models has higher correlation between model size and knowledge memory performance. However, instruction-tuning empowers the models with high-level abilities (e.g. knowledge applying tasks) and commercial models still have leading performance in general.

**Strengths:**

- The knowledge benchmark fills the blank of thoroughly evaluating world knowledge of current large language models. The taxonomy is carefully designed and rich experiments on model choices are conducted.
- The ever evolving setup has long-term benefit in considering the generalization problem in knowledge-intensive tasks.
- The self-contrast metric has a good motivation in balancing the hallucination in knowledge-based generation.
- The annotation team is of strong educational background.

**Weaknesses:**

- The knowledge-wise strength of Rouge-L used in Eq. 3 doesn't look to be strong enough in capturing the knowledge association especially when T is a free-generation result, maybe replacing the measurement with another model (e.g. a entailment model) would be better? (w/ additional computational cost)

**Questions:**

Comments
- The caption of the figures should include necessary details to understand the them if the space allows (e.g. Figure 4).

---

> ### Author Response · Authors · 2023-11-16
>
> We appreciate your careful review and constructive points. Your assessment stating that our 'knowledge benchmark fills the blank of thoroughly evaluating world knowledge of current large language models' is highly encouraging to us. According to your comments, we have edited the manuscript as your suggestions as follows:
>
> - *The knowledge-wise strength of Rouge-L used in Eq. 3 doesn't look to be strong enough in capturing the knowledge association especially when T is a free-generation result, maybe replacing the measurement with another model (e.g. a entailment model) would be better? (w/ additional computational cost)*
>
> Employing models for automatic evaluation of knowledge aspects is indeed a valuable suggestion and represents a promising direction. In the design phase of KoLA, we considered such an approach. However, we ultimately chose model-free evaluation metrics to potentially prevent the evaluation model from favoring certain types of models. In the early experiments, we tried various metrics, including some model-based metrics like BertScore, BLEU, etc., and test whether their results align with human evaluations. Bases on these, we finally choose Rouge-L. We plan to continue exploring the integration of your suggested model-based evaluation methods in subsequent seasons. Thank you once again for your insightful recommendation!
>
> - *The caption of the figures should include necessary details to understand the them if the space allows (e.g. Figure 4).*
>
> We appreciate the reviewer's suggestion to elaborate on this detail. Although we mentioned this part in the first paragraph of Appendix B.4, we overlooked emphasizing this information in the caption of Figure 4. Following your reminder, we have *enriched the captions of all the tables and figures in the paper*, further enhancing the readability of the article.
>
> **[Final Remark]** We thank you again for investing the time and effort to review our paper and for the helpful comments that helped us improve the submission. We hope that our responses will address your concerns on the evaluation metrics and caption details and sincerely invite you to engage with us if you have more questions.

---

> > ### Comment · Reviewer_VBLJ · 2023-11-22
> >
> > Thanks the authors for the reply and update the presentation of the manuscript.

---

### Official Review · Reviewer_1m8m · 2023-11-08

**Soundness:** 4 excellent
**Presentation:** 4 excellent
**Contribution:** 3 good
**Rating:** 8
**Confidence:** 4

**Summary:**

The paper presents KoLA, an interesting Knowledge-oriented LLM Assessment benchmark. It assesses LLMs on four-level knowledge-related abilities: knowledge memorization, knowledge understanding, knowledge applying and knowledge creating, with known and evolving data sources.

The authors plan to make available their data, leaderboard, participation information, and supporting tools upon acceptance. They plan to host  a new competition season every three months, updating their evolving data sources, inviting participations from both open and commercial LLMs. The paper reports analysis of two searsons run comparing 28 open-source and commercial LLMs.

I found the framework to be very interesting and insightful. The community will benefit from such a large scale analysis over knowledge-related abilities with known and evolving sources.  The paper is very well written.

**Strengths:**

The tools and data from the paper will be released upon acceptance.

The community will benefit from such a large scale analysis over knowledge-related abilities with known and evolving sources. The presented analysis is already very insightful.

The breakdown of the task using knowledge-related abilities with known and evolving sources is compelling to assess LLMs evolving capabilities.

**Weaknesses:**

I don’t see any major weaknesses in the paper.

One could argue that it is just an analysis paper, some of the insights that were drawn here might not be novel. But I feel that the presented framework will be valuable to the community. The authors have done a very good job explaining the framework in detail.

**Questions:**

I would have liked to see the human evaluation results in the main part of the paper. It would strengthen the analysis.

Do authors discuss the costs involved with these studies for both seasons?

---

> ### Author Response · Authors · 2023-11-16
>
> Thank you for your valuable feedback. We are greatly delighted to note your recognition of the contributions our paper makes in terms of data, evaluation design, and results analysis. We are happy to address the questions you’ve raised.
>
> - *I would have liked to see the human evaluation results in the main part of the paper. It would strengthen the analysis.*
>
> Adding the human evaluation to the main text is indeed a good suggestion. However, due to the constraints of length, we attempted but failed to include the results of human evaluation and its background in the content. In our revised paper, we have further highlighted this part (in Section 3), with the aim of directing readers' attention to the corresponding appendix.
>
> - *Do authors discuss the costs involved with these studies for both seasons?*
>
> This is a valuable concern. To sustain the project over the long term, we have kept the budget for each season below USD 2,000. Generally, each season’s expenditure comprises two main parts: 1) the cost of data annotation, and 2) the expenses for model deployment and API calls.
>
> The data annotation includes labeling knowledge triples and event arguments. As introduced in Appendix B, to reduce the difficulty and cost of annotation, we have pre-labeled the collected text automatically, allowing annotators to focus on the core tasks. Overall, each season requires approximately USD 660-700 for triple annotation and USD 400-420 for event annotation.
>
> Due to the scale of the KoLA test sets, the costs for model deployment and API calls have been kept within an acceptable range. The deployment expenses, estimated from GPU usage time, are approximately USD 200-240, while the API incurs an additional cost of USD 150-180.
>
> Overall, for the two seasons completed thus far, the costs have not exceeded USD 1,600 per season. With the anticipated increase in the number of models in future seasons, we believe that a budget of USD 2,000 per season should be sufficient to maintain the normal operation of the project.
>
> **[Final Remark]** Thank you once again for recognizing our work and offering invaluable suggestions for improvement. Please feel invited to leave more comments in case you have additional questions.

---

### Meta-Review · Area_Chair_k36V · 2023-12-13

**Metareview:**

The manuscript introduces KoLA, a benchmark for assessing knowledge-oriented LLMs. It outlines a taxonomy of cognitive abilities intended to yield diagnostic outcomes. The benchmark incorporates both established and emerging data sources to enhance fairness and employs contrastive metrics for broader relevance. The initial iteration of KoLA evaluates 28 different LLMs, both open-source and commercial. The findings reveal trends such as a correlation between model size and knowledge retention.

The reviewers identified several strengths including the novelty of constructing benchmark with emerging corpora, evaluating model's knowledge creation capability, and the extensiveness of the experiments.
Some reviewers remained a bit skeptical about the motivation of the benchmark, some of the evaluation metrics like ROUGE-L and some additional clarity issues.

**Justification For Why Not Higher Score:**

The motivation for the new benchmark and the insights it provides are slightly limited.

**Justification For Why Not Lower Score:**

The reviewers generally liked the work; their primary concerns were centered on the need for stronger motivation and deeper insights. Other noted weaknesses predominantly related to issues of clarity, which seem easy to address.

---

### Decision · Program_Chairs · 2024-01-16

Accept (poster)